# Tropical and mid-latitude teleconnections interacting with the Indian summer monsoon rainfall: A Theory-Guided Causal Effect Network approach

Giorgia Di Capua[1,2], Marlene Kretschmer[1], Reik V. Donner[1,3], Bart van den Hurk[2,4], Ramesh Vellore[5], Raghavan Krishnan[5], and Dim Coumou[1,2]

[1]Potsdam Institute for Climate Impact Research, Potsdam, Germany
[2]VU University of Amsterdam, Institute for Environmental Studies, Amsterdam, Netherlands
[3]Magdeburg-Stendal University of Applied Sciences, Magdeburg, Germany
[4]Deltares, Delft, Netherlands
[5]Indian Institute for Tropical Meteorology, Pune, India

*Correspondence to*: Giorgia Di Capua (dicapua@pik-potsdam.de)

**Abstract.**

The alternation of active and break phases in the Indian summer monsoon (ISM) rainfall at intraseasonal timescales characterizes each ISM season. Both tropical and mid-latitude drivers influence this intraseasonal ISM variability. The circumglobal teleconnection observed in boreal summer drives intraseasonal variability across the mid-latitudes and a two-way interaction between the ISM and the circumglobal teleconnection pattern has been hypothesized. We use causal discovery algorithms to test the ISM-circumglobal teleconnection hypothesis in a causal framework. A robust causal link from the circumglobal teleconnection pattern and the North Atlantic region to ISM rainfall is identified and we estimate the normalized causal effect (CE) of this link to be about 0.2 (a one standard deviation shift in the circumglobal teleconnection causes a 0.2 standard deviation shift in the ISM rainfall one week later). The ISM rainfall feeds back on the circumglobal teleconnection pattern, however weakly. Moreover, we identify a negative feedback between strong updraft located over India and the Bay of Bengal and the ISM rainfall acting at a biweekly timescale, with enhanced ISM rainfall following strong updraft by one week. This mechanism is possibly related to the Boreal Summer Intraseasonal Oscillation. The updraft has the strongest CE of 0.5, while the Madden-Julian Oscillation variability has a CE of 0.2-0.3. Our results show that most of the ISM variability on weekly timescales comes from these tropical drivers, though the mid-latitude teleconnection also exerts a substantial influence. Identifying these local and remote drivers paves the way for improved subseasonal forecasts.

## 1 Introduction

The Indian summer monsoon (ISM) is crucial for the Indian society, which receives 75% of its total annual rainfall during the summer months June through September (JJAS). The ISM rainfall variability at intraseasonal timescales is characterized by periods of enhanced and reduced rainfall activity over the monsoon-core region of central India. These periods are usually referred to as active (wet) and break (dry) phases, respectively, and have a duration that spans from a few days up to three weeks. Prolonged active and break spells in the ISM can lead to floods or droughts and consequently have severe socio-economic implications for the Indian subcontinent. A salient semi-permanent feature of the ISM is the "monsoon trough" (MT) which manifests as a low pressure zone extending from north-western India into the Gangetic plains and the Bay of Bengal (Rao, 1976; Krishnamurti and Sugi, 1987; Choudhury and Krishnan, 2011). The rainfall amount over the MT region is generally used to define dry and wet spells within the ISM season (Krishnan et al., 2000; Gadgil and Joseph, 2003). The position and strength of MT significantly influences the spatial distribution of monsoon precipitation and associated agricultural productivity on the Indian subcontinent, and the internal dynamics of the ISM circulation itself provides a first mechanism for intraseasonal rainfall variability (Pathak et al., 2017).

The land-sea temperature difference and the mid-tropospheric thermal forcing over the Tibetan Plateau are the prime drivers for the monsoon circulation (Yanai and Wu, 2006). Ascending motions over the Indian subcontinent enhance the northward air flux from the ocean toward the land thereby bringing moisture from the ocean and fuelling rainfall. The latent heat released by strong convective rainfall is important for sustaining the ISM circulation (Levermann et al., 2009). However, it has two opposing effects: on the one hand, the latent heat release in the early stage of an active phase enhances ascending motions by heating the mid-to-lower troposphere (Levermann et al., 2009). On the other hand, latent heat release, that propagates upward and heats the upper layers of the air column, tends to increase the static stability and inhibits further ascending motions (Saha et al., 2012). Also, rainy weather and cloudy skies can have a cooling effect on the surface, which tends to suppress convection (Krishnamurti and Bhalme, 1976). While this thermodynamic perspective is useful to understand the quasi-biweekly variations of the ISM elements locally, the spatio-temporal variations in the evolution of active and break phases over the Indian monsoon region are known to involve interactions between the wind anomalies and the northward propagation of the major rain band anomalies of the Boreal Summer Intraseasonal Oscillation (BSISO) (Wang et al., 2006; Chattopadhyay et al., 2009; Shige et al., 2017).

The Boreal Summer Intraseasonal Oscillation represents a characteristic feature of the atmospheric circulation over the northern Indian Ocean and South Asia (Goswami and Ajaya Mohan, 2001; Saha et al., 2012; Suhas et al., 2012). The BSISO is identified as a rainfall band that stretches from the Indian sub-continent to the Maritime Continent, featuring a north-west/south-east tilt and propagating north-eastward from the equatorial Indian Ocean towards South East Asia with a timescale of about one month (Wang et al., 2006; Webster and Lawrence, 2002). The BSISO can influence the oscillation between break and active phases typical for the ISM intraseasonal variability, with favorable conditions for the initiation of an active phase when the BSISO rainfall band reaches the Indian sub-continent during its northeastward propagation (Goswami and Ajaya Mohan, 2001; Suhas et al., 2012). Krishnan et al. (2000) hypothesized that the triggering of Rossby waves by suppressed convection over the Bay of Bengal initiates ISM breaks through northwest propagation of high pressure anomalies from the central Bay of Bengal into northwest India. They noted that the initiation of suppressed convection and anticyclonic anomalies over the equatorial Indian Ocean and central Bay of Bengal occurred a week prior to the commencement of a monsoon break over India followed by the traversing of suppressed anomalies from the central Bay of Bengal to northwest India in about 2-3 days.

At intraseasonal timescales, the Madden-Julian Oscillation (MJO), which governs the intraseasonal tropical climate variability operating at 30-90 day timescale, represents an important tropical driver of the ISM intraseasonal variability (Zhang, 2005). The MJO consists of a transient region of strong convective motions and enhanced precipitation, which propagates eastward from the tropical Indian Ocean to the tropical western Pacific. Normally, only one area of strong convective motions related to the MJO is present in these regions. The MJO influences the ISM system with enhanced convective rainfall activity during active MJO phases and negative rainfall anomalies during suppressed MJO phases (Anandh et al., 2018; Mishra et al., 2017; Pai et al., 2011). However, during boreal summer, the MJO strength is reduced as compared to boreal winter, and both the

MJO and the BSISO propagation and phases can be well described by the outgoing longwave radiation MJO index (OMI) (Wang et al., 2018).

Next to tropical drivers, mid-latitude circulation, the North Atlantic variability and mid-latitude wave trains have been proposed to modulate the occurrence of active and break phases of the ISM (Kripalani et al., 1997; Ding and Wang, 2005, 2007). A circumglobal teleconnection pattern, characterized by a wave number 5 has been associated with seasonal and monthly rainfall and surface air temperature anomalies across the Northern Hemisphere in summer (Ding and Wang, 2005). One way to identify this circumglobal teleconnection pattern is via point-correlation maps of the 200 hPa geopotential height with a location directly east of the Caspian Sea. A two-way interaction between ISM circulation system and the circumglobal teleconnection pattern has been hypothesized: while the diabatic heat sources associated with ISM convection can reinforce the circumglobal teleconnection pattern propagating downstream (i.e. moving from west to east following the mid-latitude westerlies), the circumglobal teleconnection pattern itself may modulate the ISM rainfall, with enhanced rainfall associated with the positive phase of the circumglobal teleconnection pattern (Ding and Wang, 2005). The circumglobal teleconnection pattern also shows interdecadal variations, with a general weakening of its major centres of action over the last three decades, which has been attributed to weakening of the ISM precipitation and to the relation of the ISM with the El Niño-Southern Oscillation (ENSO) (Wang et al., 2012). Seasonal forecast models, e.g. simulations based on seasonal forecasts from the European Centre for Medium-Range Weather Forecasts (ECMWF), tend to have difficulties reproducing this pattern correctly: the circumglobal teleconnection pattern is too weak in models and one of the possible causes could be a too weak interaction with the north-western India rainfall (Beverley et al., 2019).

The latent heat released via strong convection in the ISM region has also been shown to influence regions which are located upstream (i.e. eastward). The so-called monsoon-desert mechanism involves long Rossby waves to the west of the ISM region generated by ISM latent heating. These waves enhance the downward flow over the eastern Mediterranean and north-eastern Sahara Desert suppressing precipitation in these dry regions (Rodwell and Hoskins, 1996). A subset of CMIP5 models is able to capture this mechanism (Cherchi et al., 2014), and CMIP5 scenarios for the 21[st] century project increased ISM precipitation, despite a decrease in the strength of the ISM circulation. Thus, the monsoon-desert mechanism could contribute to drying and warming trends projected for the eastern Mediterranean region, exacerbating the desiccation conditions in these regions (Cherchi et al., 2016).

Variability of sea surface temperatures (SSTs) and mid-tropospheric variables in the North Atlantic region has been shown to influence the ISM at a wide range of timescales. At inter-decadal and interannual timescales, SSTs related to the Atlantic Multi-decadal Oscillation (AMO) index have been shown to modulate the strength of the ISM by an atmospheric bridge involving the North Atlantic Oscillation (NAO) index: positive (negative) NAO phases alter westerly winds and associated storm tracks in the North Atlantic/European area, modify tropospheric temperatures over Eurasia and thus enhance (weaken) the strength of the ISM rainfall (Goswami et al., 2006). At intraseasonal timescales, a wave train originating from the north-eastern Atlantic and propagating along an arch-shaped trajectory into central Asia may influence the intraseasonal variability

of the ISM and modulate the intensity of the northern ISM rainfall at a bi-weekly timescale, linking the latter with mid-latitude circulation features (Ding and Wang, 2007; Krishnan et al., 2009). An important feature of this wave train is the 200 hPa Central Asian High, located to the east of the Caspian Sea, i.e. over the same region used to define the circumglobal teleconnection pattern, which may trigger positive rainfall anomalies over the northern ISM region by modifying the easterly vertical shear that drives the ISM circulation and its related effect on moist dynamic instability in the ISM region. Thus, this wave train generated in the North Atlantic might aid in modulating the alternating active and break conditions over central India (Ding and Wang, 2007; Krishnan et al., 2009; Saeed et al., 2011). A positive feedback mechanism between the northern ISM and the Central Asian High has also been hypothesized, with enhanced ISM precipitation acting to reinforce the positive anomaly in the Central Asian High via a Rossby wave response related to the ISM heating source (Ding and Wang, 2007).

Lagged correlation and regression analysis have been commonly used to assess the relationship between two or more climate variables (Ding and Wang, 2005; Lau and Kim, 2011; Vellore et al., 2014). Such an analysis is useful as it gives a first information on the association of two or more variables, but it can easily lead to erroneous interpretations. For instance, two non-causally related variables can appear significantly correlated, due to strong autocorrelation and/or common driver effects (McGraw and Barnes, 2018; Runge et al., 2014). In lead-lag regression analyses, the causal direction is assumed to be from the variable that leads to the variable that lags. However, in complex dynamical systems there is often no solid basis for such assumptions. For example, linear regressions alone would suggest that surface temperatures over North and South America lead ENSO variability while the opposite causal relationship is generally accepted (McGraw and Barnes, 2018). When controlling for the autocorrelation of ENSO, this spurious correlation vanishes (McGraw and Barnes, 2018). However, due to the numerous (possible) linkages in the climate system, it is usually not obvious which variables to control for when studying the dependence of two processes.

To overcome these issues, causal discovery algorithms such as Causal Effect Networks (CEN) have been recently developed and subsequently applied to gain insights into the physical links of the climate system (Kretschmer et al., 2016, 2018; Runge et al., 2015, 2019). For a given set of time series, a CEN reconstructs the likely underlying causal structure, by iteratively testing for conditional independent relationships among the input time-series. CEN have been applied to the study of stratospheric polar vortex variability (Kretschmer et al., 2016, 2018), multi-decadal North Atlantic overturning circulation (Schleussner et al., 2014), the intraseasonal stratosphere-troposphere coupling in the Southern Hemisphere (Saggioro and Shepherd, 2019) and to study the causal interactions between the ISM, the Silk Road Pattern and the monsoon-desert mechanism (Stephan et al., 2019). Although shown to be a useful statistical tool to study teleconnection pathways, a successful application of CEN requires (such as any data-driven method) expert knowledge of the underlying physical processes, including relevant variables, time-scales and temporal resolution (Runge, 2018).

Here we study the two-way interactions of tropical and mid-latitude regions with the ISM by applying such a theory-guided causal effect analysis. First, we assess whether the connection between the ISM and the mid-latitude wave trains can be

considered causal in one or both directions. Next, we quantify the relative causal effect of tropical, mid-latitude and internal drivers on the ISM. The remainder of this paper is organized as follow: Section 2 describes the data and methods that have been applied. Section 3 presents the results obtained by applying causal discovery tools to the analysis of the ISM mid-latitude and tropical drivers. In section 4, these results are discussed in the context of the existing literature and our conclusions are presented.

## 2 Data and Methods

### 2.1 Data

We define the monsoon trough (MT) region as the region between 18°-25°N and 75°-88°E. We analyse weekly rainfall averages over these region from the CPC-NCEP (0.25°x0.25°) observational gridded global rainfall dataset over the period 1979-2016 (Chen et al., 2008) and from the Pai et al. (2015) (0.25°x0.25°) observational gridded Indian rainfall dataset over the period 1979-2017 (Pai et al., 2015). In the remainder of this paper, we will mainly focus on the results obtained for the latter data set, while those for the former are provided as parts of the Supplementary Material. Using data taken from the ERA Interim reanalysis (Dee et al., 2011) for the period 1979-2017, precursor regions are calculated from global weekly averaged gridded (1.5°x1.5°) fields including outgoing longwave radiation (OLR) at the top of the atmosphere, vertical velocity at 500 hPa (W500) and geopotential height at 200 hPa (Z200). The NAO weekly index is obtained by averaging daily data from NOAA (available at ftp://ftp.cpc.ncep.noaa.gov/cwlinks/norm.daily.nao.index.b500101.current.ascii). To identify MJO phases, we use the OLR MJO Index (OMI) provided by NOAA (https://www.esrl.noaa.gov/psd/mjo/mjoindex/). This metric features the first and second principal components obtained by the empirical orthogonal function (EOF) analysis of OLR in the tropical belt (between 30°N and 30°S) filtered to remove influences outside the MJO timescale (30-90 days). OMI PC2 corresponds to the first principal component of the Real-Time Multivariate MJO index (RMM1), which is widely used in the literature ( Wheeler and Hendon, 2004; Pai et al., 2011; Kiladis et al., 2014). Moreover, the OMI index has also been proven useful in describing the BSISO behaviour, which is relevant for the ISM break and active phases in summer (Wang et al., 2018). All time series of MT rainfall, Z200 and all datasets analysed in this work are detrended and anomalies are calculated at weekly time-steps. Thus, both the climatological and seasonal cycle are removed. Since the interannual variability may affect the analysis, we follow the approach proposed by Ding and Wang (2007) and filter the data by removing from each JJAS season its seasonal average.

### 2.2 Causal effect networks

We apply Causal Effect Networks (CEN) and the Response-Guided Causal Precursor Detection (RG-CPD) tool, two recently developed applications of the so-called Peter and Clark – Momentary Conditional Independence (PC-MCI) algorithm (Spirtes et al., 2000; Runge et al., 2014, 2019). A CEN detects and visualizes the causal relationships among a set of univariate time

series of variables (here referred to as *actors*, Kretschmer et al., 2016). The network is constructed using the PC-MCI algorithm, which is a causal discovery algorithm able to distinguish between spurious and true causal links for different time lags of interest (Runge, 2018). The term "causal" rests on several assumptions (Spirtes et al., 2000; Runge, 2018): Here, it should be understood as *causal relative to the set of analysed precursors*, meaning that the identified causal links are valid relative to the selected set of actors. Adding additional actors may change the structure of the causal network. It is therefore crucial to combine CEN with theoretical domain knowledge (i.e. our "theory-guided causal discovery analysis" approach). Other important assumptions are stationarity of time series and near-linear interactions between actors, i.e., the selected actors should have a linear behaviour at least in a first order approximation.

The PC-MCI algorithm is a two-step algorithm based on a modified version of the Peter and Clark (PC) algorithm (Spirtes et al., 2000; Runge et al., 2014, 2019). In the first step (the PC-step), the relevant conditions for each variable are identified by iterative independence testing. The following MCI-step tests whether the link between two actors can be considered causal. The false discovery rate (FDR) approach, as described by Benjamini and Hochberg, is applied to correct for inflated *p*-values due to multiple significance testing (Benjamini and Hochberg, 1995; Benjamini and Yekutieli, 2001). Each step is further described below.

In a variable set $P$ containing $n$ univariate detrended anomaly time series (the actors), the PC-step identifies the *causal parents* of each $i^{th}$ element in $P$, among all the remaining elements in $P$. First, the PC algorithm calculates plain correlations between each $i^{th}$ actor in $P$ and all the remaining elements at a certain time lag $\tau$. Those actors that significantly correlate with the $i^{th}$ actor form the set of its potential parents $P_i^0$ at lag $\tau$. Then, partial correlations between the $i^{th}$ actor and each element $j^{th}$ in $P_i^0$ (where $i \neq j$), are calculated, conditioning first on one condition, i.e., the first $k^{th}$ actor in $P_i^0$ that has the strongest correlation with the $i^{th}$ actor and $i \neq j \neq k$. If $x$, $y$ and $z$ are elements in $P$, the partial correlation between $x$ and $y$ conditioned on $z$ is calculated by first performing linear regressions of $x$ on $z$ and of $y$ on $z$ and then calculating the correlation between the residuals. If the resulting partial correlation between $x$ and $y$ is still significant at a certain significance threshold $\alpha$, $x$ and $y$ are said to be *conditionally dependent* given variable $z$, i.e., the correlation between $x$ and $y$ cannot be (exclusively) explained by the influence of variable $z$. When the opposite happens, the link is spurious and therefore filtered out, and $x$ and $y$ are called *conditionally independent*. This process leads to a reduced set of parents $P_i^1$. In the next step, the process is repeated but conditioning on two conditions, i.e. calculating the linear regression on a set of two actors, leading to a second set of parents $P_i^2$. The algorithm converges when the number of causal parents contained in $P_i^m$ is equal or greater than the number of conditions needed to calculate the next partial correlation. At the end of the PC-step, each element in $P$ has its own set of parents, which then enters the MCI-step. In the MCI-step, the partial correlation between an actor and its initial set of potential parents is calculated again but conditioned simultaneously on both the set of parents of the $i^{th}$ actor and the sets of parents of each of the parents of the $i^{th}$ actor. Those parents that pass the MCI test will then form the final set of parents for the $i^{th}$ actor (Runge et al., 2017). A numerical example of these steps is given in the SI, together with the values for each parameter used for PC-MCI. For this study, the python package TIGRAMITE version 3.0 is used (https://github.com/jakobrunge/tigramite_old).

The causal links detected via the PC-MCI algorithm are visualized in terms of a causal effect network (CEN). Each CEN is composed by circles, representing the various actors, and by arrows, with the colour indicating the strength and the arrow the direction of the detected causal links. The strength is expressed by its associated path coefficient, defined following Runge et al. (2015), as the "expected change in $X^j$ (in units of its standard deviation [s.d] and relative to the unperturbed regime) at time $t$ if $X^i$ was perturbed at time $t–\tau$ by a one s.d. delta peak". To give an example, a path coefficient of 0.5 means that a change in

the causal parent of 1 s.d. corresponds to a change in 0.5 s.d. in the analysed actor. Here, due to the fact that only lags at $\tau=1$ are accounted for, the path coefficients also correspond to the total causal effect (CE). The path coefficient of each variable on itself is here referred to as the autocorrelation path coefficient. This should not be confused with the usual definition of the lag-1 autocorrelation coefficient. The autocorrelation path coefficient is the same as the path coefficient with $i = j$, and it represents the causal influence of an actor on itself.

**2.3 Response-guided causal precursor detection**

RG-CPD identifies the causal precursors of a response variable based on multivariate gridded observational data (Kretschmer et al., 2017; Di Capua et al., 2019). It combines a response-guided detection step (Bello et al., 2015) with the PC-MCI causal discovery step (Spirtes et al., 2000; Runge et al., 2014, 2015, 2019). Without requiring an a priori definition of the possible precursors, RG-CPD searches for spatially contiguous regions in multivariate gridded data that are significantly correlated

with a variable of interest (i.e., the *response variable*) at a given lag and then detects causal precursors by filtering out spurious links due to common drivers, autocorrelation effects, or indirect links. Using correlation maps, an initial set of precursors is identified in relevant meteorological fields by finding regions in which the recorded variability precedes changes in the response variable at some lead time. Correlation values are calculated with a two-sided *p*-value for a hypothesis test whose null hypothesis is that there is no correlation, using the Wald Test with a *t*-distribution of the test statistic. All *p*-values are

corrected using the Benjamini and Hochberg false discovery rate (FDR) approach to address the variance inflation due to serial correlations (Benjamini and Hochberg, 1995; Benjamini and Yekutieli, 2001). Then, adjacent grid points with a significant correlation of the same sign at a level of α=0.05 are spatially averaged to create single one-dimensional time series, characterising the dynamics of the considered field in the so-called *precursor region* (Wilks, 2016; Willink et al., 2017). In the second step, PC-MCI identifies the set of *causal precursors* for the response variable. The results present those precursor

regions on a global map that are identified to be causally linked with the response variable (here, the MT rainfall).

# 3 Results

## 3.1 Causal testing of the two-way ISM-circumglobal teleconnection mechanism and the influence of NAO

First, we assess whether the two-way interaction between the circumglobal teleconnection pattern (that characterizes the boreal summer circulation) and the MT rainfall, as hypothesized by Ding & Wang (2005, hereafter DW2005), is reproduced using our CEN analysis. We will refer to this theory as the monsoon-circumglobal teleconnection mechanism. We also analyse the influence of the North Atlantic on the MT rainfall, as hypothesized by Ding & Wang (2007, hereafter DW2007).

Figures 1a,b show the JJAS climatology and the s.d. of weekly ISM rainfall from the Pai et al. (2015) dataset for the period 1979-2017. We average the rainfall over the MT region and identify a univariate time series, which represents the weekly variation of the ISM during JJAS over the MT region (Fig. 1c). This time series contains 18 weeks for each of the 39 analysed years, each year starting on the 27[th] of May. The analysis of the MT rainfall starts on the 3[rd] week (10[th] of June) for a total of 16 7-day time slots per year. Skipping the two first time steps (weeks) of each monsoon season is necessary since they allow detecting lagged relationships, and the PC-MCI algorithm requires to add twice the maximum time lag explored (here a maximum lag of 1 week is chosen). Information from the previous year does not interfere with the following year. The weekly timescale is considered to represent the relevant interaction between ISM rainfall and Northern Hemisphere atmospheric circulation at intraseasonal timescales (Krishnamurti and Bhalme, 1976; Ding and Wang, 2007; Vellore et al., 2014). Moreover, a weekly timescale is also adequate to represent the propagation of different BSISO phases and the switch between active and break phases of the ISM (Goswami and Ajaya Mohan, 2001).

To analyse the major mode of Z200 variability in the Northern Hemisphere mid-latitudes, following DW2005, we calculate the first and second empirical orthogonal functions (EOFs) of weekly averaged Z200 fields over the Northern Hemisphere (0°-90°N, 0°-360°E) for the summer months coinciding with the ISM season (June to September, weeks 21 to 38). Figures 2a,b show these first and second EOFs. EOF1 represents the dominant component of intraseasonal variability of the Z200 field, and qualitatively resembles the s.d. of Z200 (see Fig. S1 in the SI). EOF2 represents the second dominant pattern of intraseasonal variability of the Z200 field. However, EOF1 and EOF2 explain respectively only 6.4% and 6.2% of the intraseasonal variability, i.e., after removing mean annual cycle and interannual variability. The first 5 EOFs explain together 28.5% of the total variability. These low values are not surprising if we consider that the mean annual cycle and interannual variability have been removed, thus leaving only the disturbances from the year-specific mean state. We apply a test for pronounced separation of EOFs following North et al, (1982). This test is based on the eigenvalues of the EOF analysis and provides a rule of thumb to determine whether two EOFs are degenerate, i.e. "indistinguishable between their uncertainties"

(for further details see North et al. 1982 and Hannachi et al. 2007). The test shows that the first three EOFs are not well separated mutually, while EOF1 and EOF2 are well separated from EOF4 and EOF5 (see SI, Fig. S2). Similarly, DW2007
also found a twin EOF mode in their study, although their work focuses on the Eurasian sector only. However, by design EOFs do not necessarily reflect physical patterns (irrespective of whether the individual EOFs are separable or not). They only capture dominant patterns of variability, and here EOF2 is useful since it resembles the wave-pattern in the correlation map and, thus, likely results from the circumglobal teleconnection pattern related physics. In general, we cannot a priori exclude an influence of EOF1 on the analysed system (while an influence from EOF2 is expected). Therefore, we will test whether this
is the case in the following part of this sub-section.

DW2005 define the ISM-circumglobal teleconnection mechanism at interannual and monthly timescale, DW2007 uses daily data (after removing the interannual variability) to analyse the influence of the Eurasian wave train initiated from the North Atlantic on the north-central Indian rainfall. Thus, first we need to show that the definition of the circumglobal-teleconnection pattern as in DW2005, also applies meaningfully at weekly timescales, with the interannual variability filtered out. Figure 2c
shows the spatial correlation structure of MT rainfall with the variability of Z200 in the Northern Hemisphere at weekly timescale with Z200 leading the MT rainfall by one week (lag = -1). The arch-shaped structure which is visible in Fig. 2c over Europe and Central Asia suggests that there could be a wave pattern propagating eastward from Western Europe and affecting the MT rainfall with a 1-week lag. This hypothesis will be further tested in the next section. Following DW2005, we define a circumglobal teleconnection index (CGTI) as the weekly-mean Z200 spatially averaged over the region 35°-40°N, 60°-70°E
(white box in Fig. 2d). The contemporaneous (lag = 0) spatial correlation structure between the CGTI and Z200, i.e., the circumglobal teleconnection pattern, is shown in Fig. 2d. A large region of strong positive correlation surrounds the Caspian Sea, while downstream, a circumglobal wave train is shown with 4 positive centres of action positioned over East Asia, the North Pacific, North America and Western Europe. Despite different temporal averaging, using different datasets and the fact that we remove the interannual variability, our results with 5 centres of positive correlation strongly resemble the circumglobal
teleconnection pattern described by DW2005, in terms of sign and the geographical position of its centres of action. Thus, we conclude that the circumglobal teleconnection pattern defined by the CGTI region manifests also at weekly timescale. Moreover, the wave pattern shown over Eurasia in Fig. 2c and 2d also resembles the wave pattern identified by DW2007. Thus, this resemblance represents the first hint that the circumglobal teleconnection pattern and the Eurasian wave train analysed by DW2007 share common elements, and link the North Atlantic variability to the ISM region. In both Fig. 2c and
2d correlation values with corrected $p$-values $p < 0.05$ are highlighted with black contours.

The circumglobal teleconnection pattern (Fig. 2d) shows a very low though significant negative spatial correlation with the EOF1 pattern ($r = -0.09 \pm 0.02$; here and later, r is shown together with its confidence interval at $\alpha = 0.05$). The spatial correlation of EOF2 (Fig. 2b) with the circumglobal teleconnection pattern (Fig. 2d) over the Northern Hemisphere is $r = 0.35 \pm 0.02$, and the spatial correlation with the circumglobal teleconnection pattern increases when limited to the Eurasian sector
only (0°-90°N – 0°-150°E, $r = 0.52 \pm 0.02$). Contrarily, the spatial correlation with EOF1 and the circumglobal teleconnection is also low when only the Eurasian sector is taken into account ($r = -0.16 \pm 0.03$). When EOFs are calculated only on the

Eurasian sector, the order of the first and second EOFs is reversed, but the spatial patterns are very similar: $EOF1_{Eurasia}$ is strongly spatially correlated with EOF2 ($r = 0.89 \pm 0.01$) and with the circumglobal teleconnection pattern ($r = 0.44 \pm 0.02$; see SI, Fig. S3). Thus, the choice of the region to calculate the EOFs does not strongly affect the results. Moreover, since we

are interested in the two-way effects of the entire mid-latitude circulation and the MT rainfall, we decided to use the EOFs defined over the entire Northern Hemisphere (as shown in Figs. 2a,b).

The time series for the principal component of EOF2 also significantly correlates with the CGTI time series ($r = 0.30 \pm 0.07$) while the correlation for the first principal component is low and not significant ($r = -0.06 \pm 0.08$). The circumglobal teleconnection pattern (Fig. 2d) also strongly resembles the correlation structure between MT rainfall and Z200 at lag -1 (Fig.

2c) with a spatial correlation of $0.69 \pm 0.01$. These findings are consistent with those of DW2005, and thus illustrate a likely interaction between MT rainfall and Z200 variability.

Composite 2m-temperature and precipitation differences between weeks with strong CGTI (CGTI > 1 $CGTI_{s.d.}$ , where $CGTI_{s.d.}$ is the s.d. of the CGTI index) minus weeks with weak CGTI anomalies (CGTI < -1 $CGTI_{s.d.}$) also exhibit a clear wave train pattern originating from Western Europe and central Europe respectively. Figures 2e,f present the corresponding 2m-

temperature and precipitation anomalies from the ERA-Interim and CPC-NCEP dataset, respectively, showing anomalies with $p$-values $p < 0.05$ shaded and highlighting anomalies that have corrected $p$-values $p < 0.05$ by black contours. In both variables, a wave train from Western/Central Europe to India via European Russia is detected (in Fig. 2e,f highlighted by a black arrow). Wet and cold anomalies appear over central India and European Russia, while warm and dry anomalies are found over Western/Central Europe and east of the Caspian Sea. Warm anomalies appear together with the high and low features shown

in the circumglobal circulation pattern over Europe and central Asia (Fig. 2d), however precipitation anomalies show a slight eastward shift with respect to temperature anomalies. Precipitation anomalies are more spatially confined than those found for 2m-temperature. However, a clearly defined wave pattern appears over the Eurasian sector for both variables. Though these plots are obtained by plotting composites of weeks with CGTI > 1 $CGTI_{s.d.}$ minus composites of weeks with CGTI < -1 $CGTI_{s.d.}$, we have found that these composites behave close to linearly when plotted separately (see SI, Fig. S4). This further

supports the assumption that a linear framework is a reasonable choice in this context. Moreover, when regressing the MT rainfall on the CGTI index at different lags (from lag 0 to lag -2 weeks), the strongest relationship between the CGTI and the CPC-NCEP rainfall is found at lag -1 week, with the CGTI leading a change in MT rainfall by one week (see SI, Fig. S5). This information also further supports the choice of analysing the relationships between these two variables at lag = -1 week.

Based on the DW2005 hypothesis, we build a CEN with CGTI, MT rainfall and the principal component of EOF2 (Fig. 3a). This CEN depicts a positive two-way connection both between CGTI and MT rainfall and between CGTI and EOF2. This implies that anomalously high CGTI values (with relatively high Z200 east of the Caspian Sea) enhance MT rainfall at a 1-week lag, while lower CGTI values have the opposite effect. Note that the causal effect here is always acting in the direction of the arrow with a lag of one week, e.g., the arrow from CGTI towards MT rainfall represents a causal effect of 0.22 (given

by the colour) from CGTI to MT rainfall with a lag of one week. The link directed from MT rainfall to CGTI illustrates a

reverse influence, creating a positive feedback between CGTI and MT rainfall, supporting the monsoon-circumglobal teleconnection pattern hypothesis. The strength of the causal link between the CGTI and MT rainfall is expressed by the path coefficients (see Methods section).

The causal link strength of the CGTI acting on the MT rainfall is $\beta_{CGTI \to MT}$ ~0.2 (meaning that a change of 1 s.d. in CGTI leads to a change in 0.2 s.d in MT rainfall, while the reverse link is weaker ($\beta_{MT \to CGTI}$~0.1) but still significant. EOF2 shows a two-way link with the CGTI. The links between EOF2 and CGTI support the DW2005 hypothesis that wave trains in the mid-latitudes (represented by EOF2) affect the MT rainfall via the CGTI.

To assess whether the North Atlantic variability affects the MT rainfall (as hypothesized by DW2007), we add the NAO index to our original CEN. In order to check whether the first mode of variability in the Northern Hemisphere may also play a role in shaping MT rainfall variability, we additionally include EOF1. Figure 3b shows the resulting CEN: the causal links identified in the previous CEN (Fig. 3a) remain unaltered, and two additional positive links from NAO to CGTI and from the EOF2 to the NAO emerge. A positive NAO phase will strengthen the CGTI at 1-week lag ($\beta_{NAO \to CGTI}$~0.2), while the NAO is influenced by EOF2 ($\beta_{EOF2 \to NAO}$~0.1), though weakly. EOF1 does not show a causal connection with CGTI or with any other actor, showing that in this context, EOF1 and EOF2 present a different behaviour, though in principle not statistically separated following the North test. This CEN unveils both an influence of the mid-latitude atmospheric dynamics (EOF2) and North Atlantic variability (NAO) on MT rainfall and back, with the CGTI acting as a gateway, thus supporting both the DW2005 and DW2007 hypotheses. Substituting EOF2 with its corresponding EOF calculated over the Eurasian sector, EOF1$_{Eurasia}$, shows consistent results (see SI, Fig. S6).

**3.2 Eurasian and North Atlantic mid-latitude features affecting the Indian summer monsoon**

The EOF-based CEN analysis depicts the total hemispheric response of Z200 without differentiating among the influences of different geographical regions. To detect influential regions, we next apply the Response-guided Causal Precursors Detection (RG-CPD) scheme to search for causal precursors of both the CGTI and MT rainfall at 1-week lead time in weekly OLR and Z200 fields. In the tropical belt, OLR is often used as a proxy of rainfall and convective activity due to its relation with the temperature at the top of the clouds. Deep convection is characterized by high altitude cloud tops and low emission temperatures (and thus low OLR values), while at clear sky conditions, the emission temperature of the land surface is higher and leads to larger OLR values (Krishnan et al., 2000).

Figure 4 summarizes our corresponding findings. The right column (Fig. 4) shows the MT region and represents lag 0. Moving towards the left, the second column shows the correlation maps (top) and causal precursors (bottom) of the MT rainfall identified at 1-week lead time in Z200 (Fig. 4c). Causal precursors describe an arch-shaped wave train from Western Europe to India. The wave train features one low-pressure region (L1 over European Russia) and two high-pressure regions (the CGTI and H1 over Western Europe). The leftmost column in Fig. 4 shows the correlation maps (top) and causal precursors (bottom) of the CGTI identified in the Z200 (Fig. 4a) and OLR (Fig. 4b) fields at 1-week lead time with respect to the CGTI (2-week

lead time with respect to the MT rainfall). Again, both Z200 and OLR correlation maps show an arch-shaped wave pattern emanating from the North Atlantic and propagating towards the Caspian Sea via European Russia. The associated Z200 causal precursors for the CGTI clearly depict this mid-latitude wave train both in Z200 and OLR (Figs. 4a,b). In the Z200 field (Fig. 4a), the wave train features two lows (L1 over European Russia and L2 over the eastern North Atlantic) and two highs (the CGTI and H1 over Western Europe). OLR causal precursors (Fig. 4b) depict only the L1 and H1 components of the wave, as the correlation over L2 is not significant. Moreover, the prominent influence of the tropical belt on the CGTI is also detected (OLR1, Fig. 4b), in agreement with previous findings (se Fig. 3). This result further supports the hypothesis that a wave train coming from the mid-latitudes influences the ISM circulation system via the CGTI region as already shown in Fig. 2, while strong temperature and precipitation anomalies shown in Figs. 2e,f coincide markedly with the regions H1, L1 and CGTI identified in Figs. 4a,c. Figure 4a shows a North Atlantic pattern that resembles a negative NAO, whereas Fig. 3b indicated a positive causal link from NAO on CGTI. This seeming mismatch is explained by the difference in pressure level and lead-time with a more positive NAO pattern at lag -2 (see SI, Fig. S7).

The CEN built with MT rainfall, CGTI and the upstream part of the mid-latitude wave train, i.e., L1 and H1, is shown in Fig. 5. The causal links between the CGTI and the MT rainfall remain unaltered (see Fig. 3), with the CGTI mediating the connection between the mid-latitude wave (represented by H1 and L1) and the MT rainfall. Further, the obtained CEN can be interpreted as a wave train that propagates eastward from the East Atlantic towards the Indian monsoon region, as postulated by DW2007. Let us focus on the connections between H1 and L1. If H1 increases, then L1 decreases at 1-week lag. The backward link is positive: thus, if L1 increases H1 also increases at 1-week lag. This dampening loop is easiest explained by an eastward propagating wave. To illustrate this, we design a simple experiment using synthetic time series representing a simple cosine-wave propagating eastward with added noise sampled at two locations. We estimate the wavelength, wave speed and amplitude based on the observed properties of H1 and L1 (see SI, Fig. S8-S9). The CEN resulting from these time series is equivalent to that of H1 and L1 in Fig. 5. with a negative forward link and a positive backward link. Physically, this can be understood by a traveling wave amplifying in the forward direction, but at the same time dampening in the backward direction.

**3.3 Intraseasonal variability and tropical influence on the monsoon circulation**

Next, we perform a similar analysis applying RG-CPD to fields of vertical wind velocities (W500) and OLR to capture the internal feedbacks and dynamics of the ISM convective updraft. Their correlation maps and detected causal precursors are shown in Figs. 6a,b. MT rainfall has four causal precursors in OLR. A large region extending from the Arabian Sea towards the Maritime Continent via the Indian sub-continent (OLR1, Fig. 6a) shows a negative causal link, while another region covering parts of the Tibetan Plateau shows a positive causal link (OLR2, Fig. 6a). A third region is found north-east of the Caspian Sea, over western Russia (OLR3) with a negative causal relationship with MT rainfall. A fourth, though small, region of positive correlation is found over the equatorial Indian Ocean (OLR4, Fig. 6a). OLR1 spatially overlaps with W1, the largest causal precursor in the vertical wind field, representing the north-west/south-east tilted rainfall band related to the BSISO over

the northern Indian Ocean and western Pacific Ocean (Fig. 6b, top panel). The positive correlation of W1 and negative correlation of OLR1 with MT rainfall indicate that ascending motions and associated high-level cloud formation are followed by enhanced rainfall over the MT region with a lag of 1 week. OLR2 and OLR3 largely overlap with the regions L1 and the CGTI region identified in Fig. 4c, further supporting the importance of this mid-latitude wave pattern in modulating the rainfall over the MT region also when a different variable is selected. OLR4 hints to the presence of a seesaw of enhanced and suppressed convective activity alternating between the equatorial Indian Ocean and the Indian Peninsula, likely related to the northward propagation of the BSISO.

Figures 6c,d show two CENs constructed from the MT rainfall causal precursors in OLR and W500 (region OLR1 as defined in Fig. 6a and W1 as defined in Fig. 6b), and including CGTI and MT rainfall itself. Since the OLR and W500 fields present less organized large-scale spatial patterns than Z200, we use only the two dominant causal precursors. Although the regions OLR1 and W1 show a large spatial overlap, they represent two different components of the ISM system. OLR is calculated at the top of the atmosphere with low OLR values representing strong convective motions. W500 is calculated at 500 hPa and W1 thus represents the ascending branch of the ISM circulation cell and nearby BSISO. The CEN built with OLR1, W1 and the MT rainfall at weekly timescale (Fig. 6c) represents the intraseasonal variability of the monsoon circulation and therefore its relation with the BSISO and the initiation of active and break phases. Enhanced vertical motions (W1) precede an increase in the MT rainfall by one week; seemingly, more clouds (lower OLR1) are associated with enhanced W1 one week later (thus enhancing the ISM circulation). Contrarily, enhanced rainfall shows a negative feedback on both OLR1 and W1: stronger MT rainfall leads to reduced vertical motions and clearer skies (lower W1 and higher OLR1, respectively). Hence, this CEN depicts a negative feedback intrinsic to the ISM intraseasonal variability: while enhanced ascending motions and thus strengthened ISM circulation lead to stronger MT rainfall, enhanced rainfall leads to weaker ISM circulation and in turn diminished MT rainfall, hinting to an alternation of rainy and dry periods over the MT region, likely related to the alternation of active and break phases. Thus, from a physical point of view, this can be explained in two ways: (1) as a seesaw representing the switch of the ISM system between an active and a break phase and (2) via an increase of atmospheric static stability due to latent heat release in the higher layers of the troposphere. The latter mechanism is further analysed in Fig. S10 (see SI), where we build a CEN with MT rainfall, sea level pressure (SLP) over the Bay of Bengal (SLP_BOB), temperature at 400hPa (Tp4_MT) and temperature at 600 hPa (Tp6_MT). The resulting causal links show that while a decrease in SLP over the Bay of Bengal is followed by an increase in MT rainfall and both Tp4_MT and Tp6_MT, an increase in Tp4_MT (thus enhanced static stability) leads to a decrease in MT rainfall. The described mechanism is in agreement with what was proposed by Saha et al. (2012) and Krishnamurti and Bhalme (1976).

Next, we test the causal relationships identified between MT rainfall and W1 when adding the CGTI and MJO variability to the CEN (Fig. 6d). MJO variability is expressed by the OMI PC2 index (here referred to as MJO2), the second EOF of OLR in the tropical belt (Kiladis et al., 2014). OMI PC2 corresponds to the RMM1 index, widely used in previous work on the relationship between the MJO and the ISM (Pai et al., 2011; Kiladis et al., 2014; Mishra et al., 2017). Positive RMM1 values correspond to MJO phases 3-6 (also referred to as the active phases of MJO) and physically represent the presence of strong

convection activity propagating eastward from the Indian Ocean towards the western Pacific. Several studies show that MT rainfall is enhanced during active MJO phases (Pai et al., 2011; Mishra et al., 2017; Anandh et al., 2018). Moreover, in summer the OMI index also represents the BSISO, further linking OLR1 and W1 to the BSISO related convective activity (Wang et al., 2018). In this CEN, the link from the MT rainfall towards W1 remains unaltered (while the link in the opposite direction disappears, indicating that this link is less robust than the others), while the CGTI shows a positive feedback with W1: an increase in the CGTI causes stronger ascending motions (W1), while stronger ascending motion over the Indian region strengthens the CGTI. The CGTI has a direct positive causal link to MT rainfall (Fig. 6d). Even though the direct link from MT rainfall to CGTI has now disappeared (likely because this link was already relatively weak in a simpler CEN configuration, see Fig. 3a), the link between the ISM circulation and the mid-latitudes remains via W1. Stronger updraft over the ISM and the Maritime Continent regions (higher W1) leads to increased CGTI and vice versa as seen in Fig. 3, where higher CGTI leads to enhanced MT rainfall, both directly and via W1. MJO2 displays a positive causal link with W1, meaning that OMI PC2 positive values lead to enhanced vertical motions with 1-week lead time and, as a consequence, enhanced MT rainfall with 2-week lead time. However, stronger W1 leads to decreased MJO2 (negative causal link from W1 to MJO2). Thus, MJO2 shows exactly the same causal relationships (but opposite signs) with W1 as shown for OLR1 in Fig. 6c, likely because the OMI index is also defined based upon OLR fields in the tropical belt and the OLR pattern shown by the second EOF of the OMI index largely overlaps with our OLR1 region (Kiladis et al., 2014).

We checked how these links decay over time and found that the causal links from W1 to the CGTI and from the CGTI to the MT rainfall remain stable at lag -2 weeks, but then decay to zero at lag -3 weeks. The causal effect from W1 on the MT rainfall and from MJO2 to W1 drop to values close to zero at lag -2 and -3 (see SI, Fig. S11). This result is further visible if the timescale is changed from weekly to monthly: in this case, the spatial correlation between the circumglobal teleconnection pattern and the correlation map between MT rainfall and Z200 at lag 0 is still high (r = 0.79). However, when the correlation is calculated between MT rainfall and Z200 at lag -1 (month), the map becomes non-significant (see SI, Fig. S12). This result further suggests that these causal relationships have a characteristic timescale shorter than one month and of about 1-2 weeks. Moreover, OLR shows no significant correlation patterns with the MT rainfall and no significant causal links are detected between MT rainfall and the CGTI at monthly timescale (not shown).

### 3.4: Combining local, tropical and mid-latitude causal interactions

Finally, we bring together the findings obtained with CEN and RG-CPD throughout this study and summarize them in a single CEN to provide an overall picture and test the consistency of the results. We include the most important identified regions from both the tropics and the mid-latitudes together in a single CEN (Fig. 7) and plot the corresponding CEN over a map to help the reader to associate each actor with its corresponding approximate region (though in cases when the index is defined over all longitudes, such as EOF2, it is only possible to associate the actor with its average latitude). Specifically, this CEN is built with elements that come from both, the DW2005 and DW2007 hypotheses (Fig. 3) and from our RG-CPD analysis (Figs.

4 and 6). Moreover, to test the consistency of the results under a change of rainfall dataset, we also performed the same analysis using the CPC-NCEP rainfall dataset over the period 1979-2016 (see SI, Figs. S13-S19) and found that our results are robust when using the CPC-NCEP rainfall dataset. In Fig. 7, all causal links are reproduced with the same directions and only the magnitudes of the causal effects exhibit minor changes with respect to CEN shown in Fig. 3 and 6. Our results show that the influence of both, the mid-latitude circulation (EOF2) and the North Atlantic (NAO), on the MT rainfall is mediated via the CGTI and is robust: the structure and the direction of the causal links are retained. The backward influence of the MT rainfall on the mid-latitude circulation is weaker and more complex, as shown already in Fig. 6d. CGTI has both a direct causal link to MT rainfall and an indirect one via W1.

MT rainfall and W1 show a negative feedback (though the link from the MT rainfall towards W1 is only significant when CPC-NCEP rainfall data are used, see SI, Fig. S19), with increased W1 leading to enhanced MT rainfall but stronger MT rainfall leading to weaker W1. The MT rainfall is also influenced by MJO2 via W1 with a two-way connection. Higher MJO2 values (inked to the MJO phases 3 to 6) are followed by stronger upward motions (increased W1) one week later, which in turn causes enhanced MT rainfall two weeks later (with respect to MJO2). In the opposite direction, suppressed or weakened ascending motions promote lower MJO2 one week after, thus promoting the switch toward MJO phases 7-8 and 1-2, also known as suppressed MJO phases. Suppressed MJO phases are in turn linked with the onset of break phases of the Indian monsoon (Pai et al., 2011).

To quantify the relative influence of tropical and mid-latitude teleconnections on MT rainfall, we report the causal effect (CE) strength of each link, which is given by the path coefficient in the CEN (see Fig. 7 and Table S1 in the SI). W1 has the strongest causal effect on MT rainfall with $\beta_{W1\rightarrow MT} = 0.54$, implying that a one s.d. shift in W1 causes about a half s.d. change in MT rainfall after one week (under the previously mentioned conditions, see Methods section). The CGTI influences MT rainfall directly and indirectly via W1 and the total causal effect is given by $\beta_{CGTI\rightarrow MT} + \beta_{CGTI\rightarrow W1} \cdot \beta_{W1\rightarrow MT} = 0.23$, where $\beta_{CGTI\rightarrow MT} = 0.18$ and $\beta_{CGTI\rightarrow W1} = 0.09$ The causal effect of CGTI is thus roughly half as strong as that of the intraseasonal variability of the Indian monsoon system as represented here by W1. MJO2 has a causal effect on W1 of $\beta_{MJO2\rightarrow W1} = 0.49$ and is influenced by W1 with $\beta_{W1\rightarrow MJO2} = -0.39$. Looking more specifically at the causal effect of MJO2 on MT rainfall mediated via W1, we find $\beta_{MJO2\rightarrow W1}*\beta_{W1\rightarrow MT} = 0.26$, i.e., the tropical driver effect on MT rainfall itself is similar in magnitude as that of CGTI as for the key extratropical driver. Thus, taking both W1 and MT as representatives of ISM intraseasonal variability, the effects of external drivers from both tropics and mid-latitudes on the ISM circulation are both about half as strong as that of W1 on MT. For comparison, the path coefficients for all the identified causal links are reported together with the corresponding Pearson correlations of each pair of variables (see SI, Table S1). In general the values of the causal effect and the simple correlation do not differ greatly, despite a few exceptions where linear correlation is not significant even though there exists a causal link, or when the sign of the causal effect and the simple correlation differ. However, one should not forget that plain correlation does not indicate causality, nor can identify chains of causality between different variables.

Moreover, we calculate the average causal effect (ACE) and average causal susceptibility (ACS) for each actor. While ACE gives a measure of the causal effect that each actor has on the rest of the network, ACS measures the sensitivity of each actor

to perturbations entering in any other part of the network (Runge et al., 2015). In this CEN, W1 has the highest ACE (~0.22) and both CGTI and MJO2 the second highest ACE (~0.10). MT rainfall and W1 show the strongest ACS (~0.1). ACE and ACS values for each actor are further summarized in Table S2, see SI. These values again stress the importance of both, ISM intraseasonal variability and CGTI, in mediating mid-latitude waves towards the ISM.

Finally, we test for the robustness of the causal links when different regions other than the MT area are studied. In general, analysing all-India rainfall (AIR), western Himalayan foothills (WHF, defined over 26°-35°N and 70°-83°E) or eastern Himalayan foothills (EHF, defined over 20°-30°N and 87°-97°E) rainfall does not affect the strength or the sign of the causal links (see SI, Fig. S20-S21), thus showing that our results are robust and can actually represent the dynamics of the entire ISM basin. This is likely a consequence of the fact that AIR strongly reflect the behaviour of the MT rainfall ($r = 0.57 \pm 0.05$), as here the most abundant rainfall takes place. While the correlation between AIR and WHF rainfall is lower ($r = 0.35 \pm 0.06$), the same causal links as for the MT rainfall are observed, suggesting a strong similarity in the dynamical mechanisms that govern MT and WHF rainfall. However, when EHF rainfall is taken into account, the correlation with the AIR is low and not significant ($r = 0.03 \pm 0.05$). As a consequence, the links from the CGTI and W1 to EHF disappear (though all other links are left unchanged). This suggests that the dynamical mechanisms that bring abundant rainfall to this region are different from those that determine the MT rainfall (see SI, Fig. S21).

## 4. Discussion and conclusions

In this study, we apply causal discovery algorithms to analyse the influence of global middle and upper tropospheric fields on weekly ISM rainfall and study the two-way causal links between the mid-latitude circulation and ISM rainfall, together with tropical drivers and ISM intraseasonal variability. We perform a validation of both the monsoon-circumglobal teleconnection hypothesis proposed by DW2005 and the North Atlantic-monsoon connection proposed by DW2007 using causal discovery tools. We use the Response-guided Causal precursor detection (RG-CPD) scheme to detect causal precursors from both mid-latitude and tropical regions and then apply Causal Effect Networks (CEN) to assess the influence of different drivers of MT rainfall and their relative contribution to the MT rainfall intraseasonal variability.

The new findings of this work can be summarized in two main aspects: first, we prove the DW2005 hypothesis from a causal point of view, showing that the hypothesized two-way link between the MT rainfall and the mid-latitude circulation can be demonstrated in a causal framework. Second, we quantify the relative importance of a) the mid-latitude circulation, b) the intraseasonal variability and c) the tropical drivers of the ISM circulation, showing a link that connects the Boreal Summer Intraseasonal Oscillation (BSISO) and the Madden-Julian Oscillation (MJO) to the ISM intraseasonal variability (see Figure 7 and Table S1 in the SI). Moreover, we also show that the circumglobal teleconnection hypothesis by DW2005, initially defined at seasonal/monthly timescale, holds also at intraseasonal (weekly) timescale and it can be unified with the hypothesis by DW2007, which analysed wave trains propagating from the North Atlantic toward the MT region at 2-week timescale. Thus, we argue that the CGTI region and the mid-latitude circulation are important for both the intraseasonal and seasonal

variability of the MT rainfall. An explanation for this could be that the seasonal/monthly representation of the monsoon-circumglobal teleconnection interaction is actually strongly influenced by mechanisms that act on intraseasonal timescales, similarly to what has been proposed by Stephan et al. (2018), who come to a similar conclusion but analysing the causal interaction between the Silk Road pattern and the monsoon-desert mechanism. Here, a hint in this direction is provided by the analysis of the monsoon-circumglobal teleconnection link at monthly timescale: while the spatial correlation between the MT rainfall and the Z200 field at lag 0 strongly resembles the circumglobal teleconnection pattern (when monthly averages are analysed), this similarity disappears when Z200 leads the MT rainfall by 1 month (see SI, Fig. S12). Therefore, we suggest that the mechanism responsible for the monsoon-circumglobal teleconnection is acting at a timescale shorter than one month.

Our causal analyses confirm the influence of the mid-latitude circulation on MT rainfall via the CGTI, as hypothesized by DW2005. We also confirm that MT rainfall forces the mid-latitude circulation via the CGTI but this link is weaker (see SI, Fig. S18). We test the sensitivity of the monsoon-circumglobal teleconnection hypothesis to changes of the selected rainfall region, showing the general robustness of the identified causal links. Both all-India rainfall (AIR) and western Himalayan foothills (WHF) rainfall behave consistently with the MT rainfall (see SI, Figs. S20-21). These similarities are likely due to the strong correlation that exists between the MT rainfall and AIR, with the MT rainfall being one of the dominant features of the ISM intraseasonal variability (Krishnan et al., 2000). However, eastern Himalayan foothills (EHF) rainfall is found to behave differently, and does not show any connection with the updraft region identified by W1. This is likely related to the fact that the EHF region receives heavy rainfall amounts during the early stage of a break event and thus it is likely to be affected by different circulation patterns than those that govern the MT rainfall (Vellore et al., 2014). Nevertheless, also in this case, all other causal links in the CEN are retained.

Next to the influence of the MT rainfall, our analysis also shows that the link between the circumglobal teleconnection pattern and the ISM can be seen in the wider perspective of the BSISO variations. Our OLR1 and W1 regions (see Fig. 6a,b) show a north-west/south-east tilted rainfall band that shows great similarities with the BSISO rainfall band (Wang et al., 2018). In light of this relationship, the negative feedback that characterizes the causal links between W1 and the MJO2 (see Fig. 6d) and the MT rainfall and W1 (see Fig. S18 in the SI) can be interpreted as the shift of the ISM system between a break and an active phase and to the north-eastward propagation of BSISO convective anomalies. Therefore, while on the one hand we analyse MT rainfall (which is directly linked to active and break phases at intraseasonal timescales), on the other hand our W1 region also represents the BSISO influence on the monsoon-circumglobal teleconnection mechanism, thus linking the mid-latitude circulation not only to local latent heat release connected to the MT rainfall, but to the wider updraft region related to the South Asian monsoon in its broader definition that closely resembles the BSISO rainfall band. This connection is supported by the fact that the direct link from the MT rainfall to the CGTI disappears when W1 is added to the CEN (see Fig. 6). While an indirect link from the MT rainfall to the CGTI remains via W1, the disappearance of the direct link indicates that the influence of W1 on the CGTI is stronger than the direct link from the MT rainfall.

Applying RG-CPD, we find a wave train that emanates from the eastern North Atlantic stretching towards India via Europe and western Russia. This wave pattern is visible in geopotential height fields, temperature and precipitation anomalies, and

acts on MT rainfall via the CGTI with a 1-2 week lead time, in agreement with the DW2007 hypothesis and with previous studies showing that there is a connection from the North Atlantic toward the ISM system (Goswami et al., 2006). Moreover, the larger CGTI region as defined in Fig. 4c, while showing the strongest correlation over the CGTI region as defined in Fig. 2d, also stretches south-westwards towards North-east Africa, to the area that features the downdraft related to the monsoon-580    desert mechanism (Rodwell and Hoskins, 1996). Therefore, our work is in agreement with previous findings that show an influence of the ISM latent heat release on North-east Africa arid conditions via the excitation of Rossby waves to the west. However, here we focus on relationship between the MT rainfall and the circumglobal teleconnection pattern, thus a link from the MT rainfall towards the Saharan region cannot be inferred from this analysis. Moreover, the causal relationship between the north-eastern Indian rainfall and the downdraft over the north-eastern Sahara related to the monsoon-desert mechanism has 585    already been shown (Stephan et al., 2019).

Our results show that RG-CPD can detect well-known circulation features of the MT rainfall with 1-week lead time, without using any a priori theoretical or geographical constraint to select the causal precursors among all precursor regions demonstrating the efficacy of the presented method. Moreover, causal discovery tools can quantify the causal influence of tropical drivers versus mid-latitude influences and BSISO on the ISM intraseasonal circulation dynamics.

Intraseasonal variability of the ISM system, here represented by the updraft region identified by W1 has the strongest effect on MT rainfall. The influence from the mid-latitudes on MT rainfall, as mediated by CGTI, is about half of the magnitude of that of the internal dynamics, while the influence of MJO as the key tropical driver on MT rainfall is only slightly weaker (Fig. 7). However, when taking MT rainfall and vertical wind field over the Indian subcontinent together as two interdependent yet different facets of the ISM, we find that the general effect of tropical drivers (MJO) on the system is slightly stronger than that 595    of the extratropical drivers, while looking on the one hand on the effect of MJO on the circulation and on the other hand on the total effect of CGTI on MT rainfall via both, directed linkages and through a parallel influence on the vertical wind field over India. Though in this framework a direct influence of higher latitudes on the MT rainfall it is not identified, this may depend on the choice of the analysed (intraseasonal) timescale. However, an influence from the Arctic on the ISM rainfall has been identified at longer (inter-seasonal) timescales and has shown to provide some forecast skill for seasonal all-India rainfall 600    at 4- and 2-month lead times (Rajeevan et al., 2007; Di Capua et al., 2019;).

The reported findings are in good agreement with the existing literature. It is well known that intraseasonal variability dominates ISM inter-annual variability (Goswami and Xavier, 2005; Suhas et al., 2012). Our causal approach enables us to quantify the relative importance of local internal dynamics, separate it from the influence of remote actors, and remove spurious factors. The negative feedback in the ISM intraseasonal variability, here represented by the opposite relationship 605    between MT rainfall and W1 (see Figs. 6d and 7), features a switch from an active to a break phase inside the ISM system, likely linked to the BSISO. Other physical mechanisms can include both radiative effects (Krishnamurti and Bhalme, 1976) and local changes in static stability due to the latent heat release that follows convective precipitation (Saha et al., 2012). While strong upward motions precede strong MT rainfall, enhanced rainfall over the W1 region is known to lead the initiation of breaks by 7-10 days (Krishnan et al., 2000). Moreover, suppressed convection over the Bay of Bengal initiated

over the tropical Indian Ocean and associated westward propagating Rossby waves cause break conditions over the monsoon trough (Krishnan et al., 2000). Our results also support previous findings that suggest a link between the active MJO phases (3-6, corresponding to positive RMM1 values) and enhanced ascending motions over the MT region and adjacent Maritime Continent which in turn promote enhanced MT rainfall (Fig. 6d and 7) (Pai et al., 2011; Mishra et al., 2017; Anandh et al., 2018).

Our theory-guided causal effect network approach, i.e. creating CENs starting from physical hypotheses, enables us to: (1) test those hypotheses in a causal framework, removing the influence from spurious correlations, and (2) quantify the relative strength (i.e., the causal effect) of different local and remote actors. With this approach, one can gain insight in which role each part of a complex system such as the ISM circulation plays in relation to the other components. However, domain knowledge is essential to be guided by known physical processes and associated timescales. By combining RG-CPD and CEN,

one can test initial hypotheses and perform further more explorative causal analyses to identify new features. For example, in this study, we first identify our initial actors based on the literature. Then, we increase the pool of actors by searching for causal precursors using RG-CPD. Finally, we reconstruct a CEN that combines those findings and helps to put them into a broader context. This approach can be applied to both observational data (as done here) and climate model data to validate the underlying processes behind intraseasonal variability, which might pave the way for improved forecasts.

The described identification and quantification of causal dependences is based on linear statistical models between the different considered variables quantified in terms of partial correlations. While such linear models can provide useful approximations of real-world climate processes, there could be cases in which they miss other existing linkages that are not described by linear functional relationships. In turn, extending the present analysis to a fully nonlinear treatment is straightforward but would come on the cost of much higher computational demands, which is why we have restricted ourselves in this work to the linear

case. Nevertheless, accounting for possible nonlinearities may add further information on the inferred mechanisms and should therefore be undertaken in future research.

In conclusion, our results indicate that, on weekly timescales, the strength of the influence from the mid-latitudes on MT rainfall itself is as large as that from the tropics (MJO) but about a factor of two smaller than the ISM intraseasonal variability. However, the tropical (MJO) effect on the associated vertical wind speed over the MT region is larger than that of extratropical

drivers on MT rainfall. While previous studies that have analysed the relationships between the ISM and the mid-latitude circulation have often considered the rainfall over north-western India (Ding and Wang, 2005; Beverley et al., 2019; Stephan et al., 2019), here we take into account the MT rainfall, showing that connection between active and break phases of the ISM (by definition identified over this region, Krishnan et al., 2000) and the circumglobal teleconnection pattern. The circumglobal teleconnection pattern is an important source of variability for European summer weather, thus improving its representation in

seasonal forecasting models could in turn improve seasonal forecast in boreal summer (which generally show lower skill than those for boreal winter) (Beverley et al., 2019). Related to the confirmed relevance of extratropical drivers for ISM variability at weekly time scales, we emphasise that there exists a substantial body of literature suggesting that the influence from the

mid-latitudes is particularly important for extremes (Lau and Kim, 2011; Vellore et al., 2014, 2016). Future work should therefore aim to further disentangle the specific mechanisms that particularly act in the context of extremes.


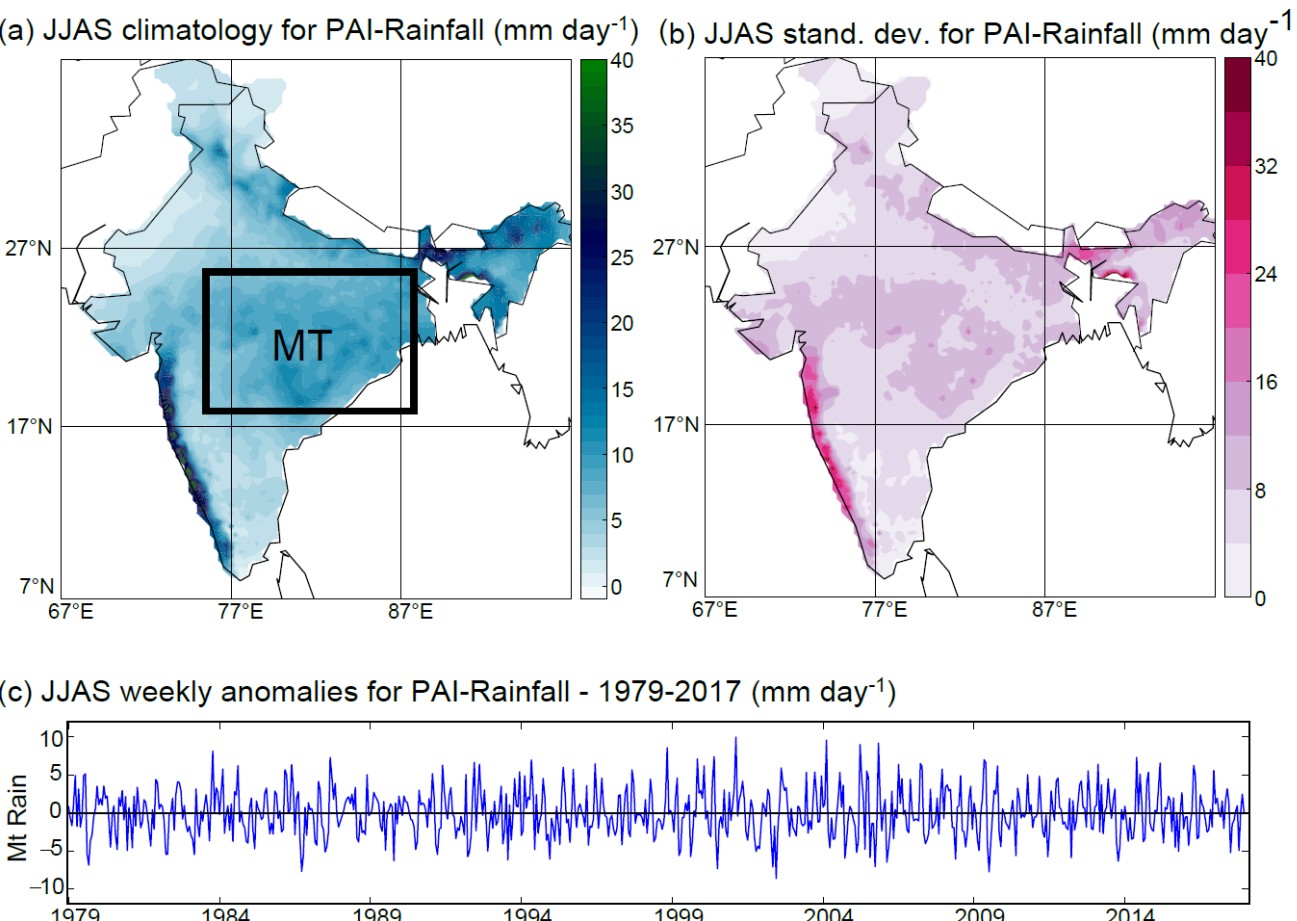

**Figure 1. Rainfall climatology over India.** Panel (a): JJAS rainfall climatology over the 1979-2017 period from the Pai et al. (2015) dataset. The black box identifies the MT region. Panel (b): standard deviation (s.d.) of weekly JJAS rainfall over the 1979-2017 period from the Pai et al. (2015) dataset. Panel (c): time series of weekly MT JJAS rainfall over the period 1979-2017; each year contains 18 weeks, with the first week starting on the 27th of May.

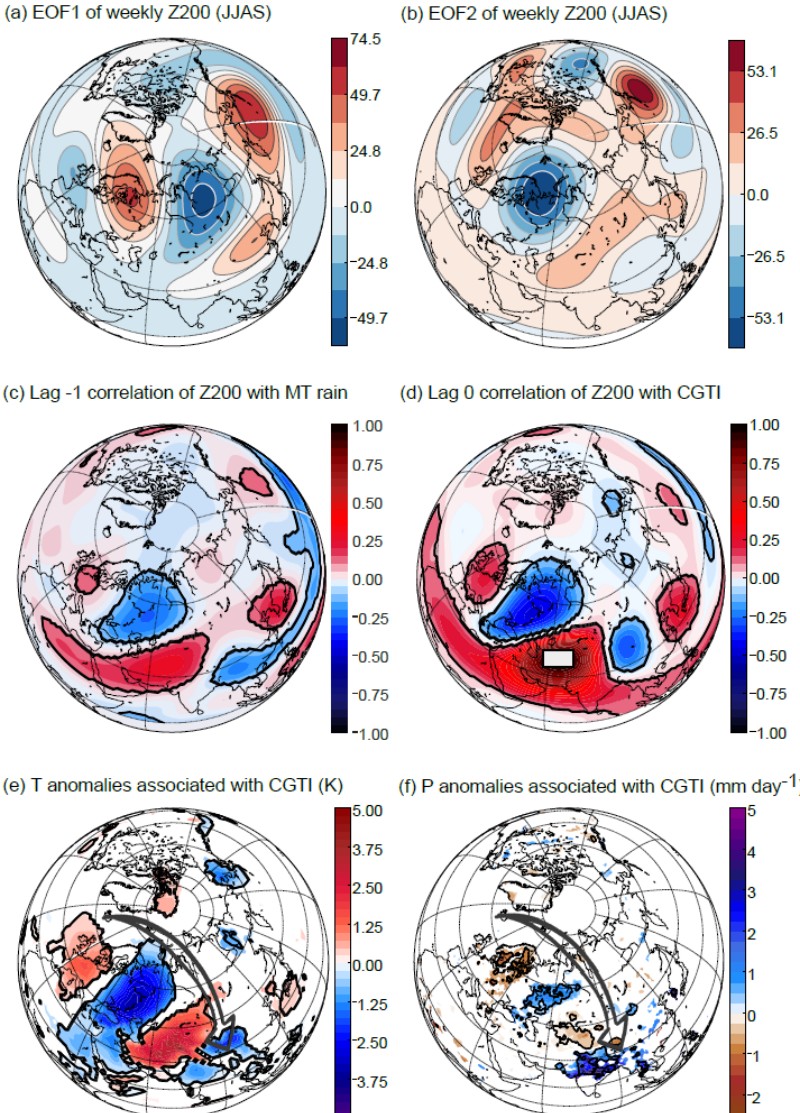

**Figure 2. Mid-latitude variability associated with the ISM.** Panels (a) and (b): EOF1 and EOF2 expressed as covariance for the JJAS weekly Z200 field in the Northern Hemisphere (0°-90°N, 0°-360°E) for the period 1979-2017. Panel (c): correlation between weekly MT rainfall (lag = 0) and Z200 (lag = -1 week). Panel (d): the CGTI region (white box) and the correlation between CGTI and Z200 (both at lag = 0), which forms the circumglobal teleconnection pattern. In panels (c) and (d), correlation coefficients with a $p$-value of $p < 0.05$ (accounting for the effect of serial correlations) are shown by black contours. Panel (e): Composite temperature difference between weeks with CGTI > 1 CGTI$_{s.d.}$ minus weeks with CGTI < -1 CGTI$_{s.d.}$ over the Northern Hemisphere. Panel (f): as panel (e) but for rainfall anomalies from the CPC-NCEP dataset (as rainfall data for the Pai et al. (2015) dataset are available over India only) for the period 1979-2017. In panels (e) and (f), anomalies with a $p$-value of $p < 0.05$ (accounting for the effect of serial correlations) are shown by black contours, while grid points which are found significant only with non-corrected $p$-values are shaded.

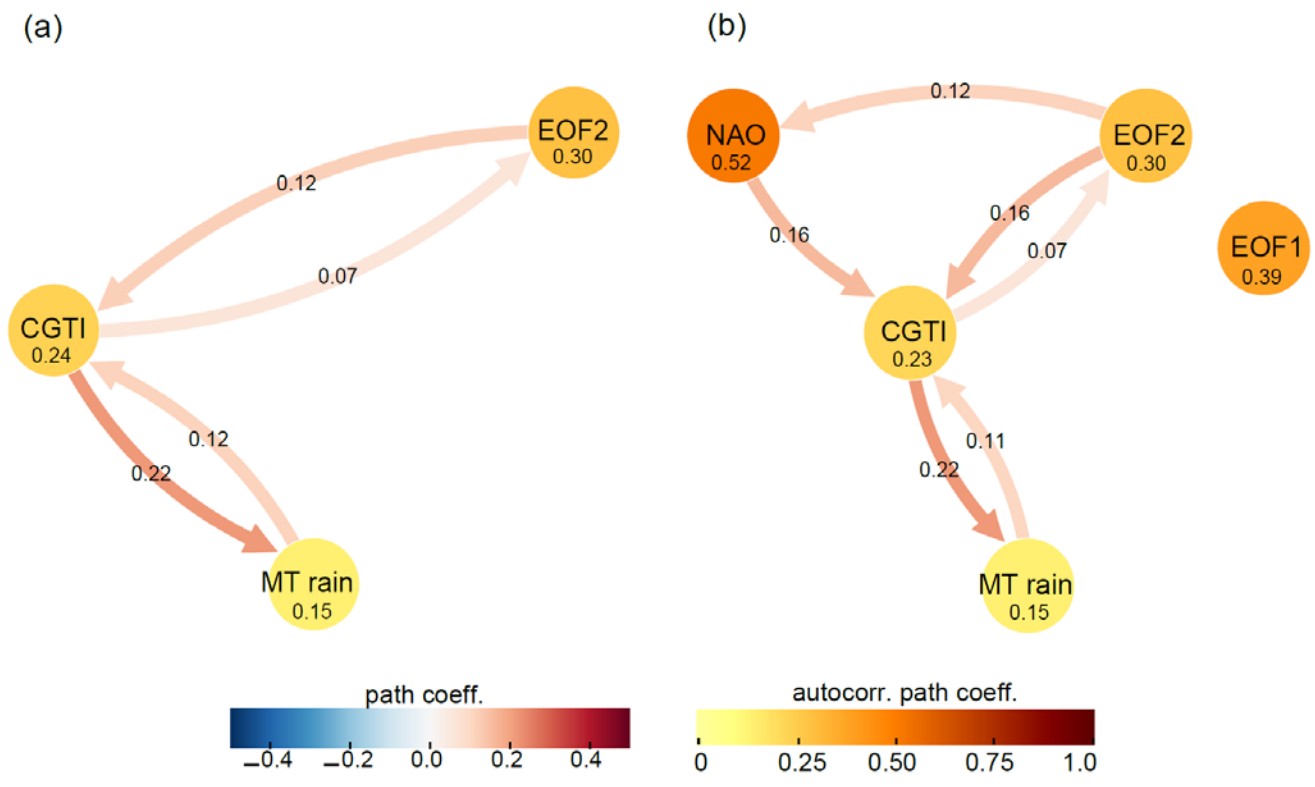

**Figure 3. Causal mid-latitude interactions with the ISM.** Panel (a): Causal Effect Network (CEN) built with CGTI, the principal component of EOF2 and MT rainfall for the period 1979-2017. Panel (b): Same as panel (a) but with the addition of the principal component of EOF1 and NAO. The strength of the causal links expressed by the standardized regression path coefficients and autocorrelation path coefficients are shown over the arrows and inside the circles respectively. All links have a lag of 1 week. Only causal links with corrected $p < 0.05$ are shown. See the main text for discussion.



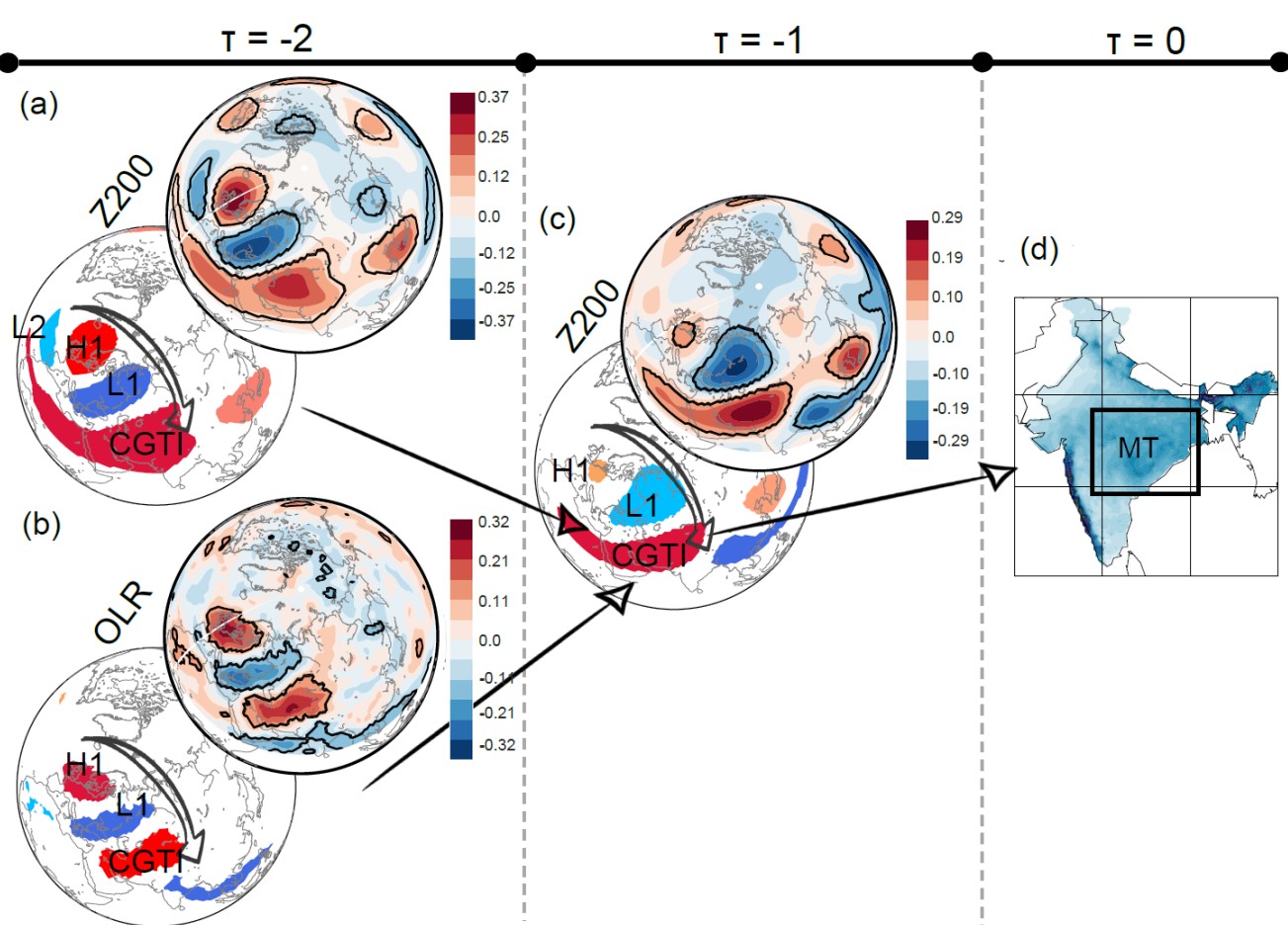

**Figure 4. Mid-latitude causal precursors of ISM.** Panel (a): correlation of CGTI with Z200 at 1-week lead time (top), and the causal precursors of CGTI identified via RG-CPD (bottom) for the period 1979-2017. Panel (b): as for panel (a) but for OLR fields. Panel (c): correlation map for weekly MT rainfall and Z200 field at 1-week lead time (top) and the causal precursors identified via RG-CPD (bottom). Panel (d): ISM rainfall over the MT region from the Pai et al. (2015) dataset (reproduced from Fig. 1a). Regions with correlation values with a *p*-value of $p < 0.05$ (accounting for the effect of serial correlations) are shown by black contours.

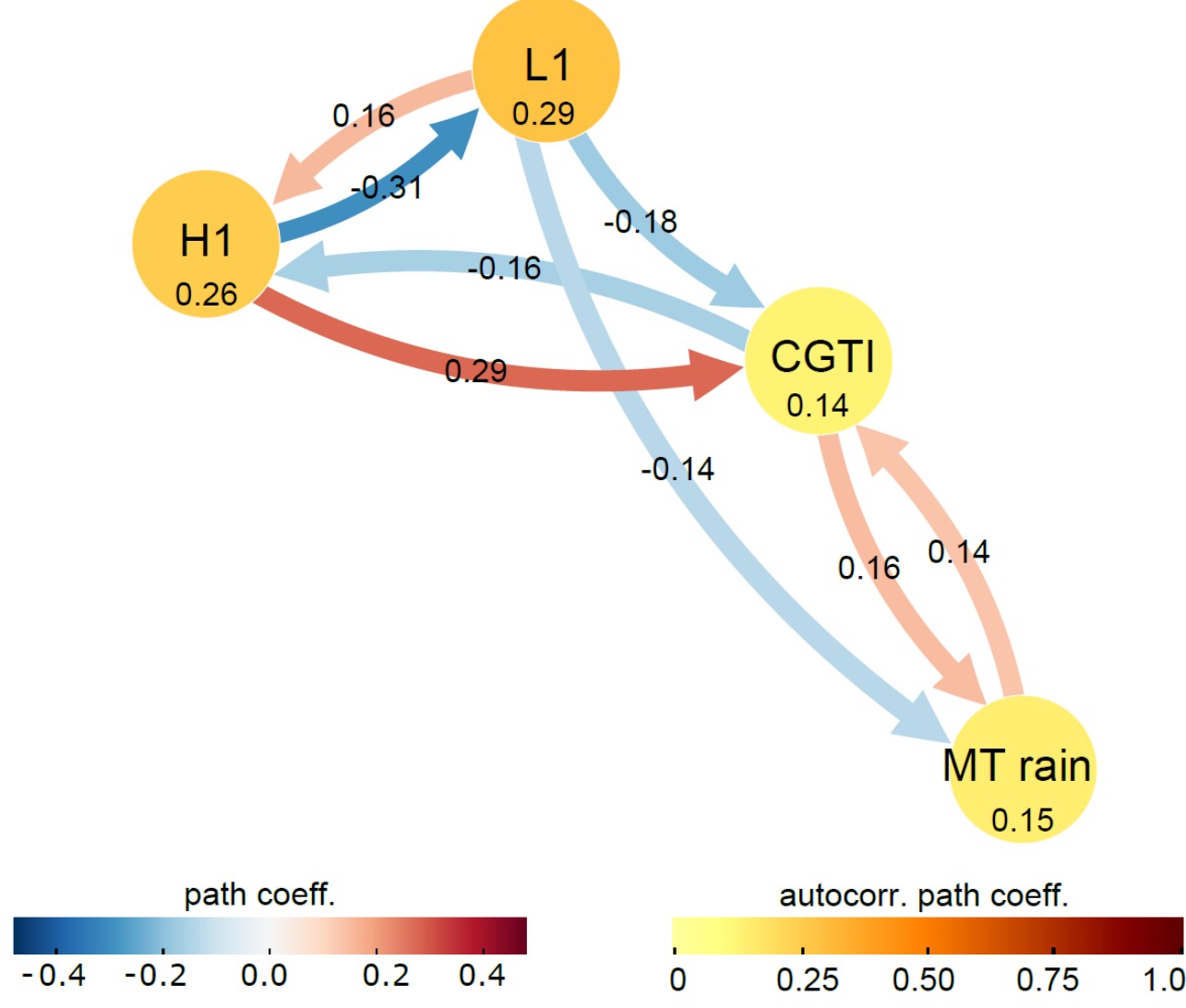

**Figure 5. Mid-latitude wave train.** CEN built with the MT rainfall, CGTI, L1 and H1 (as identified in Fig. 4a) for the period 1979-2017. The strength of causal links expressed by the standardized regression path coefficients and autocorrelation path coefficients are shown above the arrows and inside the circles respectively. All links have a lag of 1 week. Only causal links with corrected $p < 0.05$ are shown. See the main text for discussion.


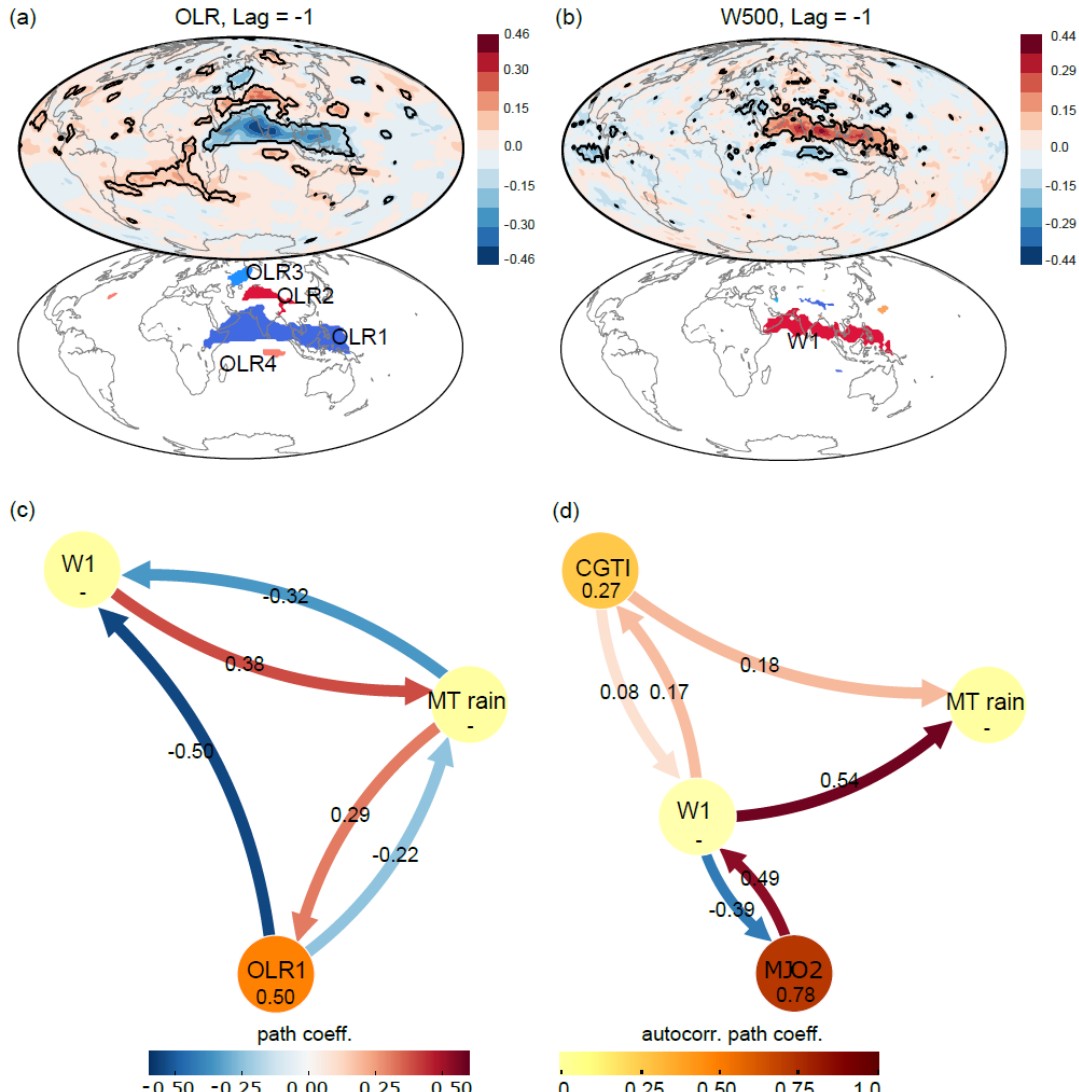

**Figure 6. Tropical causal interactions of ISM.** Panel (a) shows the correlation map of weekly MT rainfall with the global OLR field at 1-week lead time (top panel) and the causal precursors identified via RG-CPD (bottom panel) for the period 1979-2017. Regions with correlation values with a $p$-value of $p < 0.05$ (accounting for the effect of serial correlations) are shown by black contours. Panel (b): as for panel (a) but for the W500 field. Panels (c) and (d) show the CEN build with W1, OLR1 and MT rainfall and MT rainfall, W1, CGTI and MJO2, respectively. The strengths of causal links expressed by the standardized regression path coefficients and autocorrelation path coefficients are shown above the arrows and inside the circles respectively. All links have a lag of 1 week. Only causal links with corrected $p < 0.05$ are shown. See the main text for discussion.

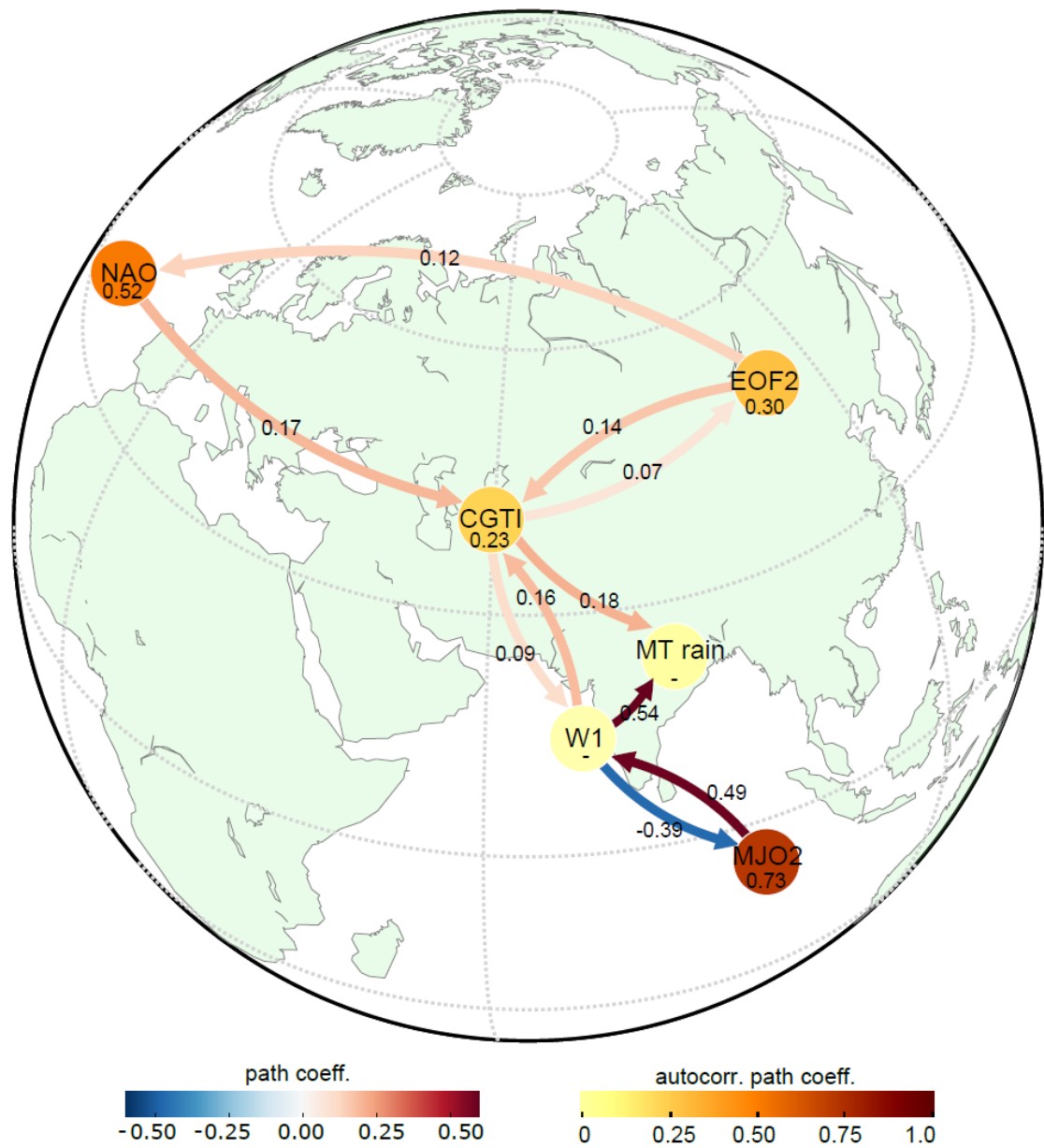

**Figure 7. Combined mid-latitude and tropical causal interactions of ISM.** CEN built with W1, MJO2, MT rainfall, NAO, CGTI and
EOF2 for the period 1979-2017. The strengths of causal links expressed by the standardized regression path coefficients and autocorrelation
path coefficients are shown above the arrows and inside the circles respectively. All links have a lag of 1 week. Only causal links with
corrected $p < 0.05$ are shown. See the main text for discussion.

## Acknowledgments

We thank ECMWF and NCEP for making the ERA-Interim and CPC-NCEP data available. G.D.C., D.C., and R.V.D. acknowledge cofunding from the German Federal Ministry of Education and Research (BMBF Grant 01LP1611A) under the auspices of the Belmont Forum and JPI-Climate project GOTHAM. R.V.D and M.K. acknowledge cofunding from the German Federal Ministry of Education and Research (BMBF Grants 01LN1306A (CoSy-CC$^2$) and 01LN1304A (Sacre-X). Code for the causal discovery algorithm is freely available as part of the Tigramite Python software package at

https://github.com/jakobrunge/tigramite. We thank the anonymous reviewers for helping us improving the content of this manuscript and for their encouraging words.

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
