# Peer review of "Tropical and mid-latitude teleconnections interacting with the Indian summer monsoon rainfall: A Theory-Guided Causal Effect Network approach"

_Earth System Dynamics, 2019_

## Referee Comment (RC1) · Anonymous Referee #1 · 13 Oct 2019

This paper tackles the important subject of teleconnections to the Indian monsoon, mainly from the midlatitudes although also with tropical elements included such as that of intraseasonal variability. Given the monsoon provides the majority of water for more than a billion people, better understanding of its teleconnections gives hope for enhancing predictions. The work uses the novel CEN technique among others to diagnose the pathways between the monsoon, the circumglobal teleconnection (Ding and Wang region) and further across Eurasia at weekly time scales. The subject matter on dynamical teleconnections is suitable for the ESD journal.

[Figure]

The paper is an important contribution to the literature and deserves to be published after some minor alterations. The paper could do more to mention previous literature such as that of Rodwell and Hoskins, which is important in explaining the mechanism for the link between the monsoon, central Asia (including the Ding and Wang region) and further afield to the Mediterranean. In addition, the treatment of what the paper describes as the internal dynamics of the monsoon is confusing. The authors should perhaps re-examine this explanation or make a judgement as to whether this can be better described as part of the monsoon intraseasonal variability. Note that in boreal summer, it would be more appropriate to refer to the monsoon ISO/BSISO rather than the MJO. The paper generally has some very well produced figures. Specific comments can be found below.

Specific comments Line 35: It is not clear what is meant by the "ISM convective cell". Is this some sort of mesoscale convective system or an individual cloud? I suggest that some alternative terminology is found. (See also later on line 39.)

Line 36: "Periods with strong updraft lead to strong rainfall one week later". Please clarify if this means locally or acting at some distance along the teleconnection.

Line 37: In your statement, "internal ISM dynamics has the strongest CE of 0.5", what does this mean in the context of the similar statement earlier that explains the meaning of these value? Effectively you are saying (unless I am misinterpreting the purpose of the CE value), "A one standard deviation shift in the internal ISM dynamics causes a 0.5 standard deviation shift in ISM rainfall one week later". It is not clear what you are meaning by this.

Line 43: In the introduction it may be worth also citing the work of Stephan et al. (2019, https://doi.org/10.1175/JCLI-D-18-0405.1) who use the CEN technique to examine the CGT/SRP link to the ISM, albeit in the context of decadal variability.

Lines 63-64: In the statement, "...this thermodynamic perspective [cloud cover etc. acting to cool the surface] is useful to understand the quasi-biweekly variations of the

[Figure]

ISM elements locally", why is the quasi-biweekly time scale of particular interest? In a monsoon regime wouldn't we expect CAPE to build up and be destroyed much more regularly than this, e.g. on a daily basis, given that the surface forcing is strong and there is a plentiful supply of moisture?

Line 77: Rather like the earlier comment, what is meant by a "convective cell" of the MJO here?

Line 87: Regarding "downstream", careful to specify what is meant. Do you mean downstream with respect to the jet, i.e. further east? This seems to be in the same direction as given in the previous sentence, rather than "on the other hand" as the sentence starts.

Line 166 onwards: the methods in this section are explained well given the complex techniques involved and the referencing is done very well.

Figure 2 and others in the paper are very inventive and generally of very good quality.

Line 205: How are the "northern mid-latitudes" used for the EOF calculation defined?

Line 205: "lag = -1 week". It would be better to clarify exactly which variable is leading the other, to avoid ambiguity.

Line 224: That the first three EOFs are not mutually separable is somewhat of a mathematical interpretation, but what does it mean in terms of any physical explanation of the EOFs?

Line 232: Perhaps it would be a good idea to add a further physical interpretation of the Z200 leading MT rainfall by 1 week. Presumably this is a west to east propagation of the signal.

Line 247: Does the global scale mentioned here imply that the correlation was performed over all global gridpoints? It seems a bit excessive and could probably just be done over the hemisphere.

Line 275: Here (and also for the benefit of later), please clarify that the lags of 1-week act in the direction of the arrows (e.g. CGTI leads MT by one week, and then MT feeds back on CGTI one week later).

Line 281: Consistency with Ding and Wang is mentioned here, but ultimately this also supports Rodwell and Hoskins (1996) on monsoon-desert coupling (https://doi.org/10.1002%2Fqj.49712253408). This seems to be a surprising omission from the paper given that it helps explain the relationship between the monsoon and Ding & Wang region, and ultimately further afield to the Mediterranean. It would be good to discuss this work in the introduction and perhaps see also the works of Cherchi et al. (2014; https://doi.org/10.1175/JCLI-D-13-00530.1, which examines the issues in coupled models, and also see the discussion of the Rodwell and Hoskins mechanism in relation to Ding & Wang in Beverley et al. (2019; https://doi.org/10.1007/s00382-018-4371-4).

In figure 4, panel (d) seems rather pointless. Can a composite difference of ISM rainfall not be given based on the precursors illustrated in the earlier panels of the figure?

Line 326: What do the black contours represent in the figure?

Lines 330-332: It may be worth citing some earlier works linking NAO and monsoon, e.g. Goswami et al. (2006, https://doi.org/10.1029/2005GL024803).

Line 348: The "Himalayan plateau" is not appropriate as it does not exist. Do you mean the Himalayas or the Tibetan Plateau (or both)? Better to think of a more appropriate term.

Line 351-355: Rather than the "ITCZ", isn't an interpretation of this that the one week earlier than strong rain over the trough, we have rain further south over India, such that we have a northward propagation of the BSISO? The rainband looks slightly titled rather than entirely zonal. That would be why you see a causal relationship of MJO2 to W1.

[Figure]

Lines 369-370: There was an explanation on the use of OLR earlier, so it isn't needed again.

Line 372: Here and elsewhere we are referred to the internal dynamics of the monsoon. Is this not perhaps better explained as the intraseasonal variability, in other words related to the BSISO of active and break phases?

Lines 373-374: The stronger MT rainfall followed by weaker ascending motions one week later is probably consistent with a monsoon active phase moving into a break.

Line 377-380: The negative feedback described here is logical in the sense that convective rainfall acts to stabilize the atmospheric column (and destroy CAPE). But in the monsoon regime, CAPE will quickly be reinvigorated given the surface forcing and good availability of moisture, possibly within a day. Rather than a negative feedback, couldn't one also argue that (as in Gill's off-equatorial heating), the LH release of the monsoon convection leads to a feedback and strengthening of the flow. What you are seeing here may instead be better explained as part of the migration between states of the BSISO.

Lines 454-455: Are the internal variability and the MJO-related part not somehow related, i.e. they are intraseasonal variability of the monsoon.

Lines 461-462: Monsoon-desert coupling could be mentioned here.

Line 478: The description here of internal variability dominating over interannual variability might be better explained in terms of intraseasonal versus interannual.

Lines 485-486: It would have been useful to have these discussions about the ITCZ migration earlier. This might be better explained in any case in terms of the switch between break and active phases of the BSISO, as the region of maximum rainfall propagates northward from the equatorial position during a break to Indian latitudes during an active phase.

Line 489: Here and elsewhere, the MJO is discussed but it may be better to think in

terms of the BSISO, which is the summer manifestation of the MJO with northward propagation.

Line 499 (and elsewhere): It would be better to expand acronyms such as RG-CPD or explain again what they are in the conclusions, for the benefit of the reader that goes straight to the conclusion section.

Spelling, grammar & other trivia Line 34: Change "influences back" to "feeds back on" Line 63: "in support of suppressing" is rather contradictory English. I suggest changing it to something like, "which tends to suppress convection". Line 76: Insert "The" before "MJO". Line 103: It would be customary for a few sentences here at the end of the introduction listing what sections are to follow in the remainder of the paper. Line 125: missing space before "algorithm". Line 188 and elsewhere: Perhaps put the year of the Pai et al. reference here and elsewhere. Line 208: In describing panel (e) it would be clearer to express this as a composite difference, e.g., "Composite temperature difference between weeks with...". Line 232: timescale –> timescales Line 240: patter –> pattern Line 243: llink –> link Line 262: As in the earlier comment, it might be best to explain as a composite difference. Line 318: "figure" for the new sentence should be capitalized. Line 479: Enables to –> enables us.

---

## Referee Comment (RC2) · Anonymous Referee #2 · 16 Oct 2019

The paper addresses an interesting topic from a network point of view, confirming previous results with this new data-driven learning methodology. However, I think that the authors should make more clear which is the added value of this methodology (with respect to using more traditional methodologies) and highlight which results are different from what was already known. In particular, the determination of the causality of the links seems to be the most important strength of the method, but this is not sufficiently stated. In addition, I find that the selection of some parameters (weekly time-span, CGTI region, 1 week time-lag, precipitation only over MT and not over all

the Monsoon area, etc.) are poorly justified and some discussion on how the results are modified with a different selection is necessary. Finally, some word on why the linear framework is adequate for the study if these mechanisms is missing.

Major questions/comments:

1. Which is the added value of the methodology with respect of using just correlations and partial correlations? I think that probably the values of the links could be reproduced just with more simple statistics but the causality determination is the highlight of the proposed methodology. This should be more clearly emphasized and showing the values of the correlations / partial correlations among all the selected variables (MT precipitation, MJO2, PC2, CGTI, etc.) is advised for comparison.

2. Which is the time-decay of the normalized causal effect among the different links? The authors only mention the results for 1-week lag but I think it's interesting to comment on how the intensity f the connections decays (or does not) with time.

3. How do the results change when seasonal or monthly time-scales are selected? As mentioned, the original DW2005 hypothesis was originally defined for the seasonal/monthly time-scales bu this time-scales are avoided in the current manuscript. Why? How do the results change?

4. The DW2005 is the base for this study, however it is only briefly discussed. I suggest to include a whole paragraph of the Introduction to discuss their findings more deeply. Also, a deeper discussion of DW2007 is missed. Those 2 studies are mentioned but together with other studies and, thus, their relevance and main results are difficult to identify.

5. The authors focus on mid-latitude and tropical links to the Indian Mosnoon. Can the authors identify any high-latitude link with the MT precipitation?

6. Why is only the MT region selected? Which are the results when selecting, for example the maximum precipitation over western India or eastern of 87E? How about

selecting the precipitation over all the Indian Monsoon region?

7. In all the network figures it would be useful to have the numbers indicating the path coefficient and auto-corr. path coefficient over the arrows and inside the circles, respectively. Absence of arrows indicates 0 path. Coefficient?

8. Why linearity is a good framework here? Please cite works to justify this.

Minor comments:

The terms "mid-latitudes" and "extra-tropics" are used indistinctly. This is problematic and in "extra-tropics" the high-latitudes are also included. Please only use 1 term to avoid confusing the reader.

line 125: space missing ")algorithm"

line 129: space missing ":It should"

line 130: what is an actor? A variable?

line 133: what is near-linear? Define.

line 134: PC algorithm means PC-MICI algorithm or other? Also the name is confusing as you used the term for Principal component before

line 149: which are the 2 conditions? More, generally: which are the n-conditions?

line 150: "parents contained in $P^{n}_i$" is n the same as before? I guess not, change for m

line 161: why is only tau=1 selected? Justify, why not look at other taus?

Line 176-180: why is the correction needed?

Line 183: how is "circumglobal wave train" defined?

Line 186: why is the NAO influence included in this subsection? The title only talks about ISM and circumglobal wave train

fig 1: panel c: add in the title that it's over the MT region. Y-axis should be 10 to 10. Xaxis: it would be easier if it indicates the years

line 197: why start on the 2rd week? Justify and only show fig 1c starting at this time.

Line 199: justify why max lag 1 week is selected

line 243: typo "llink"

line 248-249: why the uncertainty in r? isn't it just the pattern correlation number? Please explain.

Line 246-255: re-arrange paragraph to talk first about EOF1 and then about EOF2. Also, mention definition of eurasian sector the first time it appers.

Line 261-266: in Precipitation there is no signal over western Europe

line 269: Careful! How can you compare variables of different magnitude????!! you can't say the precipitation is weaker than temperature.

Fig 2: CGTI has not been defined up to this point in the text. Why not show als lag + -1? seems important later. Why show temperature anomalies when it's not the focus of the paper? Panels e and f: subtitles are misleading, indicate it's anomalies associated with extreme CGTI.

Line 227-241: AT this pont CGTI seems important to the paper, however it is not shown. I suggest to include its time-series. Also a justification on why such a small region is selected is needed.

Line 316: OLR1 is not indicated in the figure 4b section 3.3: why add MJO in this section when it is about internal feedbacks? The sub-title is misleading

fig 7: why is the network overlapped on a lon-lat map? No overlapping was done before. For example, why is EOF2 located over east Asia or W1 over west India?

Line 446-447: you only mention mid-latitudes even though later tropics and internal

feedbacks are analysed.

Line 459: how does your results have implications for the interannual time-scales? I think this sentence is misleading.

Line 464-465: how does this pattern compare with the regression of MT precipitation on the Z200 field? Any substantial difference?

Line 486-488: is it possible to confirm this with your results?

Lines 502-508: why is the linear framework adequate for studying these mechanisms? Can you cite any modelling work implying linearity of these type of interactions?

———————————————

---

## Author Comment (AC1) · 12 Nov 2019

Response to review #1 The paper is an important contribution to the literature and deserves to be published after some minor alterations. The paper could do more to mention previous literature such as that of Rodwell and Hoskins, which is important in explaining the mechanism for the link between the monsoon, central Asia (including the Ding and Wang region) and further afield to the Mediterranean. In addition, the treatment of what the paper describes as the internal dynamics of the monsoon is confusing. The authors should perhaps re-examine this explanation or make a judgement as to whether this can be better described as part of the monsoon intraseasonal variability. Note that in boreal summer, it would be more appropriate to refer to the monsoon ISO/BSISO rather than the MJO. The paper generally has some very well produced figures. Specific comments can be found below.

- Answer -

We thank the anonymous reviewer for his/her very positive overall feedback and useful suggestions on missing literature, on how to better discuss and compare the existing literature in the context of our work and on how to improve the clarity of the paper. We have included all suggestions in our revised version of the paper, with a particular attention on discussing the analogies and differences that our results show if compared to previous work. We now describe the BSISO in the introduction and discussion section, with a more specific attention to its relation to MJO in summer. We have included the suggested literature in the introduction and we now compare our results in light of previous contributions to this specific topic. A fully detailed answer to each comment is reported below.

Specific comments

Line 35: It is not clear what is meant by the "ISM convective cell". Is this some sort of mesoscale convective system or an individual cloud? I suggest that some alternative terminology is found. (See also later on line 39.) – Answer – We thank the reviewer for pointing out this unclear terminology. It is indeed more appropriate to substitute the term "convective cell" with the more general term "circulation system" (of the ISM). We have thus made this substitution in the text.

Line 36: "Periods with strong updraft lead to strong rainfall one week later". Please clarify if this means locally or acting at some distance along the teleconnection. – Answer – We have further specified the location where the updraft is located, see lines 36-37: "Moreover, we identify a negative feedback between strong updraft located over India and the Bay of Bengal and the ISM rainfall acting at a biweekly timescale, with

enhanced ISM rainfall following strong updraft by one week.".

Line 37: In your statement, "internal ISM dynamics has the strongest CE of 0.5", what does this mean in the context of the similar statement earlier that explains the meaning of these value? Effectively you are saying (unless I am misinterpreting the purpose of the CE value), "A one standard deviation shift in the internal ISM dynamics causes a 0.5 standard deviation shift in ISM rainfall one week later". It is not clear what you are meaning by this. – Answer – The meaning of the CE is correctly understood. We have however specified that here by internal dynamics it was actually meant the updraft identified over the Indian subcontinent and the Bay of Bengal. The new sentence now reads (lines 35-38) "Moreover, we identify a negative feedback between strong updraft located over India and the Bay of Bengal and the ISM rainfall acting at a bi-weekly timescale, with enhanced ISM rainfall following strong updraft by one week. This mechanism is possibly related to the Boreal Summer Intra-seasonal Oscillation. In our analyses, the updraft has the strongest CE of 0.5, while the Madden-Julian Oscillation variability has a CE of 0.2-0.3.".

Line 43: In the introduction it may be worth also citing the work of Stephan et al. (2019, https://doi.org/10.1175/JCLI-D-18-0405.1) who use the CEN technique to examine the CGT/SRP link to the ISM, albeit in the context of decadal variability. – Answer – We thank the anonymous reviewer for suggesting to include this work in our paper. We now refer to this reference in the introduction to highlight how CENs can be applied to study this type of problem, see line 148-151: "CENs can be used to test whether hypothesized links or teleconnections are likely to represent true physical pathways or rather artefacts due to spurious correlations. CEN have been applied to the study stratospheric polar vortex variability (Kretschmer et al., 2016), multi-decadal North Atlantic overturning circulation (Schleussner et al., 2014) and to study the causal interactions between the ISM, the Silk Road Pattern and the monsoon-desert mechanism (Stephan et al., 2019), showing their usefulness in testing existing hypothesis by eliminating those that are not supported by causality ."". We discuss these results more

in detail in the discussion section, and compare their outcomes with our analysis, see lines 624-629: "Therefore, our work is in agreement with previous findings that show an influence of the ISM latent heat release on north-east Africa arid conditions via the excitation of Rossby waves to the west. However, here we focus on the relationship between the MT rainfall and the circumglobal teleconnection pattern, thus a link from the MT rainfall towards the Saharan region cannot be inferred from this analysis. Moreover, the causal relationship between the north-eastern Indian rainfall and the downdraft over the north-eastern Sahara related to the monsoon-desert mechanism has already been shown (Stephan et al., 2019).".

Lines 63-64: In the statement, ". . .this thermodynamic perspective [cloud cover etc. acting to cool the surface] is useful to understand the quasi-biweekly variations of the ISM elements locally", why is the quasi-biweekly time scale of particular interest? In a monsoon regime wouldn't we expect CAPE to build up and be destroyed much more regularly than this, e.g. on a daily basis, given that the surface forcing is strong and there is a plentiful supply of moisture? Answer - We agree. Following the also the comments below, we now introduce here the BSISO, and refer to the above described mechanism as a potential local driver of the ISM intraseasonal variability. See lines 64-68: "While this thermodynamic perspective is useful to understand the quasi-biweekly variations of the ISM elements locally, the spatio-temporal variations in the evolution of active and break phases over the Indian monsoon region are known to involve interactions between the wind anomalies and the northward propagation of the major rain band anomalies of the Boreal Summer Intraseasonal Oscillation (Chattopadhyay et al., 2009; Shige et al., 2017; Wang et al., 2006). ".

Line 77: Rather like the earlier comment, what is meant by a "convective cell" of the MJO here? – Answer – Corrected sentence lines 84-85: "Normally, only one region of strong convective motions related to the MJO is present in these regions.".

Line 87: Regarding "downstream", careful to specify what is meant. Do you mean downstream with respect to the jet, i.e. further east? This seems to be in the same

direction as given in the previous sentence, rather than "on the other hand" as the sentence starts. – Answer – We thank the anonymous reviewer for carefully reading the manuscript. We have changed the structure of the paragraph and we now discuss separately the DW2005 and DW2007 related mechanisms in lines 90-107 and 116-131 respectively, this should improve the readability of the paper.

Line 166 onwards: the methods in this section are explained well given the complex techniques involved and the referencing is done very well.

Figure 2 and others in the paper are very inventive and generally of very good quality.

Line 205: How are the "northern mid-latitudes" used for the EOF calculation defined? – Answer – We have clarified this point as follows (see lines 268-269):"Panels (a) and (b): EOF1 and EOF2 expressed as covariance for the JJAS weekly Z200 field in the Northern Hemisphere (0°-90°N, 0°-360°E) for the period 1979-2017.".

Line 205: "lag = -1 week". It would be better to clarify exactly which variable is leading the other, to avoid ambiguity. – Answer - We have clarified this point as follows (see lines 269-271): "Panel (c): correlation between weekly MT rainfall (lag = 0) and Z200 (lag = -1 week). Panel (d): the CGTI region (white box) and the correlation between CGTI and Z200 (both at lag = 0), which forms the circumglobal teleconnection pattern.".

Line 224: That the first three EOFs are not mutually separable is somewhat of a mathematical interpretation, but what does it mean in terms of any physical explanation of the EOFs? – Answer - We have clarified this point as follows (lines 287-292): "However, by design EOFs do not necessarily reflect physical patterns (irrespective of whether the individual EOFs are separable or not). They only capture dominant patterns of variability, and here EOF2 is useful since it resembles the wave-pattern in the correlation map and thus it likely results from the circumglobal teleconnection pattern related physics. In general, we cannot a priori exclude an influence of EOF1 on the analysed system (while an influence from EOF2 is expected), therefore we will test whether this is the case in the following part of this sub-section.".

Line 232: Perhaps it would be a good idea to add a further physical interpretation of the Z200 leading MT rainfall by 1 week. Presumably this is a west to east propagation of the signal. – Answer – we have included this comment as follows (lines 298-300): "The arch-shaped structure which is visible in Fig. 2c over Europe and central Asia suggests that there could be a wave pattern propagating eastward from Western Europe and affecting the MT rainfall with a 1-week lag. This hypothesis will be further tested in the next section.".

Line 247: Does the global scale mentioned here imply that the correlation was performed over all global grid-points? It seems a bit excessive and could probably just be done over the hemisphere. – Answer – We thank the anonymous reviewer for this remark, we did mean that the spatial correlation and the EOFs are calculated only over the Northern Hemisphere and we have corrected this ambiguity (see lines 277-279).

Line 275: Here (and also for the benefit of later), please clarify that the lags of 1-week act in the direction of the arrows (e.g. CGTI leads MT by one week, and then MT feeds back on CGTI one week later). – Answer – We have clarified this point including the text the following sentence, (see lines 356-358) :"Note that the causal effect here is always acting in the direction of the arrow with a lag of one week, e.g. the arrow from CGTI towards MT rainfall represents a causal effect of xx (given by the color) from CGTI to MT rainfall with a lag of one week.".

Line 281: Consistency with Ding and Wang is mentioned here, but ultimately this also supports Rodwell and Hoskins (1996) on monsoon-desert coupling (https://doi.org/10.1002%2Fqj.49712253408). This seems to be a surprising omission from the paper given that it helps explain the relationship between the monsoon and Ding & Wang region, and ultimately further afield to the Mediterranean. It would be good to discuss this work in the introduction and perhaps see also the works of Cherchi et al. (2014; https://doi.org/10.1175/JCLI-D-13-00530.1, which examines the issues in coupled models, and also see the discussion of the Rodwell and Hoskins mechanism in relation to Ding & Wang in Beverley et al. (2019; https://doi.org/10.1007/s00382-

018-4371-4). – Answer – We thank the anonymous reviewer for suggesting to include the discussion of the monsoon-desert mechanisms in our work, we now discuss all the suggested references in the introduction, lines 108-115 "The latent heat released via strong convection in the ISM region has also been shown to influence regions which are located upstream (i.e. eastward). The so-called monsoon-desert mechanism involves long Rossby waves to the west of the ISM region generated by ISM latent heating. These waves enhance the downward flow over the eastern Mediterranean and north-eastern Sahara Desert suppressing precipitation in these dry regions (Rodwell and Hoskins, 1996). A subset of CMIP5 models is able to capture this mechanism (Cherchi et al., 2014) and CMIP5 scenarios for the 21st century project increased ISM precipitation, despite a decrease in the strength of the ISM circulation. Thus the monsoon-desert mechanism could contribute to drying and warming trends projected for the eastern Mediterranean region, exacerbating the desiccation conditions in these regions (Cherchi et al., 2016)" and lines 103-107 "Seasonal forecast models, e.g. from the European Centre for Medium-Range Weather Forecasts (ECMWF), tend to have difficulties reproducing this pattern correctly: the circumglobal teleconnection pattern is too weak in models and one of the possible causes could be a too weak interaction with the north-western India rainfall (Beverley et al., 2019). " as well as in the discussion on the paper lines 622-625 "Moreover, the larger CGTI region as defined in Fig. 4c, while showing the strongest correlation over the CGTI region as defined in Fig. 2d, also stretches south-westwards towards north-east Africa, to the area that features the downdraft related to the monsoon-desert mechanism (Rodwell and Hoskins, 1996). Therefore, our work is in agreement with previous findings that show an influence of the ISM latent heat release on north-eastern Africa arid conditions via the excitation of Rossby waves to the west." and lines 679-685 "While previous studies that have analysed the relationships between the ISM and the mid-latitude circulation have often considered the rainfall over north-western India (Beverley et al., 2019; Ding and Wang, 2005; Stephan et al., 2019), here we take into account the MT rainfall, showing that connection between active and break phases of the ISM (by definition

identified over this region, Krishnan et al., 2000) and the circumglobal teleconnection pattern. The circumglobal teleconnection pattern is an important source of variability for European summer weather, thus improving its representation in seasonal forecasting models could in turn improve seasonal forecast in boreal summer (which generally show lower skill than those for boreal winter) (Beverley et al., 2019).".

In figure 4, panel (d) seems rather pointless. Can a composite difference of ISM rainfall not be given based on the precursors illustrated in the earlier panels of the figure? – Answer – The purpose of this panel in Fig. 4 is only to help the interpretation, as it helps to visualize which region is acting on each response variable at any given lag. lags

Line 326: What do the black contours represent in the figure? – Answer – We have clarified this point as follows for both Fig. 4 and Fig. 6 (lines 275 and 409): "Regions with correlation values with a p-value of $p < 0.05$ (accounting for the effect of serial correlations) are shown by black contours.".

Lines 330-332: It may be worth citing some earlier works linking NAO and monsoon, e.g. Goswami et al. (2006, https://doi.org/10.1029/2005GL024803). – Answer – We thank the anonymous reviewer for suggesting this useful reference. We now cite this work in the introduction, see lines 117-121: "At inter-decadal and interannual timescales, SSTs related to the Atlantic Multi-decadal Oscillation (AMO) index have been shown to modulate the strength of the ISM by an atmospheric bridge involving the North Atlantic Oscillation (NAO) index: positive (negative) NAO phases modify westerly winds and associated storm tracks in the North Atlantic/European area, and modify tropospheric temperatures over Eurasia, thus enhancing (weakening) strength of the ISM rainfall (Goswami et al., 2006).". We have also included this reference in the discussion section, see lines 619-621: "This wave pattern is visible in geopotential height fields, temperature and precipitation anomalies, and acts on MT rainfall via the CGTI with a 1-2 week lead time, in agreement with the DW2007 hypothesis and with previous studies showing that there is a connection from the North Atlantic toward the

ISM system (Goswami et al., 2006).".

Line 348: The "Himalayan plateau" is not appropriate as it does not exist. Do you mean the Himalayas or the Tibetan Plateau (or both)? Better to think of a more appropriate term. – Answer – We thank the anonymous reviewer for noticing this mistake. We indeed meant the Tibetan Plateau and we have now corrected this mistake (see line 432).

Line 351-355: Rather than the "ITCZ", isn't an interpretation of this that the one week earlier than strong rain over the trough, we have rain further south over India, such that we have a northward propagation of the BSISO? The rainband looks slightly titled rather than entirely zonal. That would be why you see a causal relationship of MJO2 to W1. – Answer – We thank the anonymous reviewer for this insightful suggestion. We know refer to the BSISO instead of to the ITCZ, see lines 434-436: "OLR1 spatially overlaps with W1, the largest causal precursor in the vertical wind field, representing the north-west/south-east tilted rainfall band related to the BSISO over the northern Indian Ocean and western Pacific Ocean (Fig. 6b, top panel).".

Lines 369-370: There was an explanation on the use of OLR earlier, so it isn't needed again. – Answer – We thank the anonymous reviewer for pointing out this repetition, which we have now removed.

Line 372: Here and elsewhere we are referred to the internal dynamics of the monsoon. Is this not perhaps better explained as the intraseasonal variability, in other words related to the BSISO of active and break phases? – Answer – We thank the anonymous reviewer for this insightful suggestion. We know refer to the interannual variability and the BSISO instead, see lines 458-460: "The CEN built with OLR1, W1 and the MT rainfall at weekly timescale (Fig. 6c) represents the intraseasonal variability of the monsoon cell and its relation with the BSISO and the initiation of active and break phases. ".

Lines 373-374: The stronger MT rainfall followed by weaker ascending motions one

week later is probably consistent with a monsoon active phase moving into a break. – Answer – We have now included this suggestion in our manuscript, see lines 466-469: "Thus, from a physical point of view, this can be explained in two ways: (1) as a seesaw representing the switch of the ISM system between an active and a break phase and (2) via an increase of atmospheric static stability due to latent heat release in the higher layers of the troposphere."

Line 377-380: The negative feedback described here is logical in the sense that convective rainfall acts to stabilize the atmospheric column (and destroy CAPE). But in the monsoon regime, CAPE will quickly be reinvigorated given the surface forcing and good availability of moisture, possibly within a day. Rather than a negative feedback, couldn't one also argue that (as in Gill's off-equatorial heating), the LH release of the monsoon convection leads to a feedback and strengthening of the flow. What you are seeing here may instead be better explained as part of the migration between states of the BSISO. – Answer – We thank the reviewer for this insightful comment, here (and later) we now discuss the BSISO as a possible explanation of the presence of a negative feedback between W1 and the MT rainfall, see lines 357-360: "W is calculated at 500 hPa and W1 thus represents the ascending branch of the ISM circulation cell and nearby BSISO. The CEN built with OLR1, W1 and the MT rainfall at weekly timescale (Fig. 6c) represents the intraseasonal variability of the monsoon circulation and therefore its relation with the BSISO and the initiation of active and break phases.".

Lines 454-455: Are the internal variability and the MJO-related part not somehow related, i.e. they are intraseasonal variability of the monsoon. – Answer – We thank the reviewer for this insightful comment, here (and later) we now discuss the intraseasonal variability instead of the internal dynamics, see lines 470-472: "In this study, we apply causal discovery algorithms to analyse the influence of global middle and upper tropospheric fields on the ISM rainfall and study the two-way causal links between the mid-latitude circulation and ISM rainfall, together with tropical drivers and ISM intraseasonal variability.".

Lines 461-462: Monsoon-desert coupling could be mentioned here. – Answer – We thank the reviewer for this useful suggestion, we now include the monsoon desert mechanisms in the discussion section, see lines 620-623: "Moreover, the larger CGTI region as defined in Fig. 4c, while showing the strongest correlation over the CGTI region as defined in Fig. 2d, stretches south-westwards towards north-east Africa, to the area that features the downdraft related to the monsoon-desert mechanism (Rodwell and Hoskins, 1996; Stephan et al., 2019). ".

Line 478: The description here of internal variability dominating over interannual variability might be better explained in terms of intraseasonal versus interannual. – Answer – We have included this suggestion in lines 644-626: "The reported findings are in good agreement with the existing literature. It is well known that internal intraseasonal variability dominates ISM inter-annual variability (Goswami and Xavier, 2005; Suhas et al., 2012).".

Lines 485-486: It would have been useful to have these discussions about the ITCZ migration earlier. This might be better explained in any case in terms of the switch between break and active phases of the BSISO, as the region of maximum rainfall propagates northward from the equatorial position during a break to Indian latitudes during an active phase. – Answer – We have included this suggestion and we now discuss the BSISO link to the MT rainfall already in lines 605-610 "Next to the influence of the MT rainfall, our analysis also shows that the link between the circumglobal teleconnection pattern and the ISM can be seen in the wider perspective of the BSISO variations. Our OLR1 and W1 regions (see Fig. 6a,b) show a north-west/south-east tilted rainfall band that shows great similarities with the BSISO rainfall band (Wang et al., 2018). In light of this relationship, the negative feedback that characterizes the causal links between W1 and the MJO2 (see Fig. 6d) and the MT rainfall and W1 (see Fig S18, in the SI) can be interpreted as the shift of the ISM system between a break and an active phase and to the north-eastward propagation of BSISO convective anomalies.".

Line 489: Here and elsewhere, the MJO is discussed but it may be better to think

in terms of the BSISO, which is the summer manifestation of the MJO with northward propagation. Answer – We have included this suggestion throughout the whole manuscript and we now discuss the BSISO together with the MJO. Moreover, we also discuss evidence that the OMI index we use to describe the MJO, is also useful to describe the BSISO, see lines 87-89 "However, during boreal summer, the MJO strength is reduced as compared to boreal winter, and both the MJO and the BSISO propagation and phases can be well described by the outgoing longwave radiation MJO index (OMI) (Wang et al., 2018).". In the discussion section, we now also include the discussion of the BSISO next to the MJO, see previous comments for an example.

Line 499 (and elsewhere): It would be better to expand acronyms such as RG-CPD or explain again what they are in the conclusions, for the benefit of the reader that goes straight to the conclusion section. – Answer – We have included this suggestion in the discussion and conclusion section of the paper.

Spelling, grammar & other trivia Line 34: Change "influences back" to "feeds back on" – Answer – We thank the reviewer for carefully reading the manuscript, we have now included this suggestion in the paper. Line 63: "in support of suppressing" is rather contradictory English. I suggest changing it to something like, "which tends to suppress convection". – Answer – We thank the reviewer for carefully reading the manuscript, we have now included this suggestion in the paper. Line 76: Insert "The" before "MJO". – Answer – We thank the reviewer for carefully reading the manuscript, we have now included this suggestion in the paper. Line 103: It would be customary for a few sentences here at the end of the introduction listing what sections are to follow in the remainder of the paper. – Answer – We thank the reviewer for carefully reading the manuscript, we have now included this suggestion in the paper. Line 125: missing space before "algorithm". – Answer – We thank the reviewer for carefully reading the manuscript, we have now included this suggestion in the paper. Line 188 and elsewhere: Perhaps put the year of the Pai et al. reference here and elsewhere. – Answer – We thank the reviewer for carefully reading the manuscript, we have now included

this suggestion in the paper. Line 208: In describing panel (e) it would be clearer to express this as a composite difference, e.g., "Composite temperature difference between weeks with...". – Answer – We thank the reviewer for carefully reading the manuscript, we have now included this suggestion in the paper. Line 232: timescale –> timescales – Answer – We thank the reviewer for carefully reading the manuscript, we have now included this suggestion in the paper. Line 240: patter –> pattern – Answer – We thank the reviewer for carefully reading the manuscript, we have now included this suggestion in the paper. Line 243: llink –> link – Answer – We thank the reviewer for carefully reading the manuscript, we have now included this suggestion in the paper. Line 262: As in the earlier comment, it might be best to explain as a composite difference. – Answer – We thank the reviewer for carefully reading the manuscript, we have now included this suggestion in the paper. Line 318: "figure" for the new sentence should be capitalized. – Answer – We thank the reviewer for carefully reading the manuscript, we have now included this suggestion in the paper. Line 479: Enables to –> enables us. – Answer – We thank the reviewer for carefully reading the manuscript, we have now included this suggestion in the paper.

---

## Author Comment (AC2) · 12 Nov 2019

Response to review #2

The paper addresses an interesting topic from a network point of view, confirming previous results with this new data-driven learning methodology. However, I think that the authors should make more clear which is the added value of this methodology (with respect to using more traditional methodologies) and highlight which results are different from what was already known. In particular, the determination of the causality

of the links seems to be the most important strength of the method, but this is not sufficiently stated. In addition, I find that the selection of some parameters (weekly time-span, CGTI region, 1 week time-lag, precipitation only over MT and not over all the Monsoon area, etc.) are poorly justified and some discussion on how the results are modified with a different selection is necessary. Finally, some word on why the linear framework is adequate for the study if these mechanisms is missing. –

Answer –

We thank the anonymous reviewer for his/her suggestions to improve the clarity of the paper. We have taken all suggestions into account and we now provide a clearer explanation on why causal tools generate better interpretable results as compared more basic statistical approaches like correlation. We also address the issues on the linear framework and show that the results are robust when different regions are selected. A detailed answer to each point is provided below. Please refer to the attachment for the new Figures cited below. Major questions/comments:

1. Which is the added value of the methodology with respect of using just correlations and partial correlations? I think that probably the values of the links could be reproduced just with more simple statistics but the causality determination is the highlight of the proposed methodology. This should be more clearly emphasized and showing the values of the correlations / partial correlations among all the selected variables (MT precipitation, MJO2, PC2, CGTI, etc.) is advised for comparison. -

Answer – We thank the reviewer for pointing out that the description of the advantages of using causal discovery tools instead of simple correlation needs to be emphasized more. As a clarification, the causal algorithm we use is based on partial correlation, calculated on the residuals obtained after regressing two selected variables on a set of third variables (conditions). This is a lengthy iterative process whereby one conditions on all possible variable combinations for each individual link. We explain this in full length in the SI of the paper and shorter in the main text in lines 141-158. Following the suggestions of the reviewer, we have added some clarifying examples, and expanded the paragraph that describes the advantages of using the causal effect over simple correlation, see introduction section, lines 132-151: "Lagged correlation and regression analysis are commonly used to assess the relationship between two or more climate variables (Ding and Wang, 2005; Lau and Kim, 2011; Vellore et al., 2014). Such an analysis is useful as it gives a first information on the association of two or more variables, but it can easily lead to erroneous interpretations. For instance, two non-causally related variables can be significantly correlated, due to strong autocorrelation and common driver effects (McGraw and Barnes, 2018; Runge et al., 2014). In lead-lag regression analyses, often the causal direction is assumed to be from the variable that leads to the variable that lags. However, in complex dynamical systems there is no solid basis for such assumptions. For example, linear regressions alone would suggest that surface temperatures over North and South America lead ENSO variability while the opposite causal relationship is generally accepted (McGraw and Barnes, 2018). When controlling for the autocorrelation of ENSO, this spurious correlation vanishes (McGraw and Barnes, 2018). However, due to the numerous (possible) linkages in the climate system, it is usually not obvious for which variables to control for when studying the dependence of two processes. To overcome these issues, causal discovery algorithms such as Causal Effect Networks (CEN) have been recently developed and subsequently applied to gain insights into the physical links of the climate system (Kretschmer et al., 2016, 2018; Runge et al., 2015a). For a given set of time-series, CEN reconstructs the likely underlying causal structure, by iteratively testing for conditional independent relationships among the input time-series. CEN have been applied to the study of stratospheric polar vortex variability (Kretschmer et al., 2016, 2018), multi-decadal North Atlantic overturning circulation (Schleussner et al., 2014) and to study the causal interactions between the ISM, the Silk Road Pattern and the monsoon-desert mechanism (Stephan et al., 2019). Although shown to be a useful statistical tool to study teleconnection pathways, a successful application of CEN requires (such as for any data-driven method), expert knowledge of the underlying physical

processes, including relevant variables, time-scales and temporal resolution (Runge, 2018). ". Further, following the reviewer's suggestion, we have added the simple correlation between the most important links for comparison, see SI, Table S1 and a relative comment in lines 559-564: "The path coefficients for all the identified causal links are reported together with the Pearson correlations of each pair of variables (see SI, Table S1). In general the values of the causal effect and the simple correlation do not differ greatly, despite a few exceptions where linear correlation is not significant even though there exists a causal link, or when the sign of the causal effect and the simple correlation differ. However, one should not forget that correlation does not give a sign for causality, nor can identify chains of causality between different variables.".

2. Which is the time-decay of the normalized causal effect among the different links? The authors only mention the results for 1-week lag but I think it's interesting to comment on how the intensity of the connections decays (or does not) with time. - Answer – We thank the reviewer for this suggestion. We have calculated the causal effect also for lags -2 and -3 and we show the results in figure S11 in the Supplementary material. We have inserted a corresponding comment to these results in the main text, see lines 494-496: "We checked how these links decay over time and found that the causal links from W1 to the CGTI and from the CGTI to the MT rainfall remain stable at lag -2 weeks, but then decay to zero at lag -3 weeks. The causal effect from W1 on the MT rainfall and from MJO2 to W1 drop to values close to zero at lag -2 and -3 (see SI, Fig. S11)."

3. How do the results change when seasonal or monthly time-scales are selected? As mentioned, the original DW2005 hypothesis was originally defined for the seasonal/monthly time-scales but this time-scales are avoided in the current manuscript. Why? How do the results change? Answer – Using seasonal (JJAS averaged) values, though in principle feasible, would not give a insightful information since in this case lag -1 (season) would go in the previous spring and thus not capture the distinct relationship that exists in boreal summer. Moreover, DW2005 define the circumglobal

teleconnection pattern based on summer and analyse link within the summer season. While the correlation map between Z200 and the CGTI region at monthly time scale remains very similar to that for weekly time scale, the correlation map between MT rain and Z200 at lag -1 month for monthly data is not qualitatively similar to the circumglobal teleconnection pattern anymore (see Fig. S11 in the SI). One reason for this behaviour could be that the characteristic time scale of detected causal links is biweekly (as shown also in fig. S11, see earlier comment) and thus the relationship disappears at monthly timescale. Moreover, bi-weekly time scale is also a typical time scale for quasi-stationary waves in boreal summer (Kornhuber et al. 2016). We have clarified these points as follow in the main text in lines 497-504: "This result is further visible if the timescale is changed from weekly to monthly: in this case, the spatial correlation between the circumglobal teleconnection pattern and the correlation map between MT rainfall and Z200 at lag 0 is still high (r = 0.79). However, when the correlation is calculated between MT rainfall and Z200 at lag -1 (month), the map becomes unsignificant (see SI, Fig. S12). This result further suggests that these causal relationships have a characteristic timescale shorter than one month and of about 1-2 weeks. Moreover, OLR shows no significant correlation patterns with the MT rainfall at lag -1 (month) and no significant causal links are detected between MT rainfall and the CGTI at monthly timescale (not shown)."

4. The DW2005 is the base for this study, however it is only briefly discussed. I suggest to include a whole paragraph of the Introduction to discuss their findings more deeply. Also, a deeper discussion of DW2007 is missed. Those 2 studies are mentioned but together with other studies and, thus, their relevance and main results are difficult to identify. Answer – We thank the reviewer for highlighting that these two references need to be described in more details in the introduction section. We now dedicate to each of them a full paragraph, see lines 90-104 "Next to tropical drivers, mid-latitude circulation, the North Atlantic variability and mid-latitude wave trains have been proposed to modulate the occurrence of active and break phases of ISM (Ding and Wang, 2005, 2007; Kripalani et al., 1997). A circumglobal teleconnection pattern, characterized by a wave number 5 has been associated with seasonal and monthly rainfall and surface air temperature anomalies across the northern hemisphere in summer (Ding and Wang, 2005). One way to identify this circumglobal teleconnection pattern is via point-correlation maps of the 200hPa geopotential height with a location directly east of the Caspian Sea. An two-way interaction between ISM circulation system and the circumglobal teleconnection pattern is hypothesized: while the diabatic heat sources associated with ISM convection can reinforce the circumglobal teleconnection pattern propagating downstream (i.e. moving from west to east following the mid-latitude westerlies), the circumglobal teleconnection pattern itself may modulate the ISM rainfall, with enhanced rainfall associated with the positive phase of the circumglobal teleconnection pattern (Ding and Wang, 2005). The circumglobal teleconnection pattern also shows interdecadal variations, with a general weakening of its major centres of action over the last three decades, which has been attributed to weakening of the ISM precipitation and to the relation of the ISM with the El Nino-Southern Oscillation (ENSO) (Wang et al., 2012). Seasonal forecast models tend to have difficulties reproducing this pattern correctly: the circumglobal teleconnection pattern is too weak in models and one of the possible causes could be a too weak interaction between the north-western India rainfall (Beverley et al., 2019)." and lines 121-131 "At intraseasonal timescales, a wave train originating from the north-eastern Atlantic and propagating with an arch-shaped trajectory into central Asia may influence the intraseasonal variability of the ISM and modulate the intensity of the northern ISM rainfall at a bi-weekly timescale, linking the latter with mid-latitude circulation features (Ding and Wang, 2007; Krishnan et al., 2009). An important feature of this wave train is the 200 hPa central Asian High, located to the east of the Caspian Sea, i.e. over the same region used to define the circumglobal teleconnection pattern, which may trigger positive rainfall anomalies over the northern ISM region by modifying the easterly vertical shear that drives the ISM circulation and its related effect on moist dynamic instability in the ISM region. Thus, this wave train generated in the North Atlantic might aid in modulating the alternating active and break conditions over central India (Ding and Wang, 2007; Krishnan et al.,

2009; Saeed et al., 2011). A positive feedback mechanism between the northern ISM and the central Asia High is also hypothesized, with enhanced ISM precipitation acting to reinforce the positive anomaly in the central Asia High via a Rossby wave response related to the ISM heating source (Ding and Wang, 2007)." in the main text.

5. The authors focus on mid-latitude and tropical links to the Indian Monsoon. Can the authors identify any high-latitude link with the MT precipitation? Answer – We thank the anonymous reviewer for bringing up this topic. The proposed analysis does not highlight any influence of high–latitudes on the MT rainfall. However, this result is likely a consequence relatively short time scale of the analyses. Some studies indicate that Arctic regions can have an influence on all-India rainfall on seasonal timescales. A statement addressing this topic is available in lines 641-644: "Though in this framework a direct influence of higher latitudes on the MT rainfall it is not identified, this may depend on the choice of the analysed (sub-seasonal) time scale. However, an influence from the Arctic on the ISM rainfall has been identified at longer (interseasonal) timescales and has shown to provide some forecast skill for seasonal all-India rainfall at 4- and 2-month lead times (Rajeevan, 2007; Di Capua et al., 2019)."

6. Why is only the MT region selected? Which are the results when selecting, for example the maximum precipitation over western India or eastern of 87E? How about selecting the precipitation over all the Indian Monsoon region? - Answer – We choose the MT regions as in this area defines the breaks and active phases of the ISM (see introduction, lines 48-55). However, following the reviewer's suggestion, we now provide in the SI an analogous figure as Fig. 7 in the main text but for all-India rainfall (AIR), eastern (EHF) and western Himalayan foothills (WHF) rainfall (see Fig. S20 and S21 in the SI). Results are commented in lines 558-568: "Finally, we test for the robustness of the causal links when different regions other than the MT area are selected. In general, analysing all-India rainfall (AIR), western Himalayan foothills (WHF, defined over 26°-35°N and 70°-83°E) or eastern Himalayan foothills (EHF, defined over 20°-30°N and 87°-97°E) rainfall does not affect the strength or the sign of the causal links (see

[Figure]

SI, Fig. S20-S21), thus showing that our results are robust and can actually represent the dynamics of the entire ISM basin. This is likely a consequence of the fact that AIR strongly reflect the behaviour of the MT rainfall (r = 0.57 ± 0.05), as here the most abundant rainfall takes place. While the correlation between AIR and WHF rainfall is lower (r = 0.35 ± 0.06), the same causal links as for the MT rainfall are observed, suggesting a strong similarity in the dynamical mechanisms that govern MT and WHF rainfall. However, when EHF rainfall is taken into account, the correlation with the AIR is low and not significant (r = 0.03 ± 0.05). As a consequence, the links from the CGTI and W1 to EHF disappear (though all other links are left unchanged). This suggests that the dynamical mechanisms that bring abundant rainfall to this region are different than those that determine the MT rainfall (see SI, Fig. S21).". This findings are further discussed in lines 596-604 in the discussion section "We test the sensitivity of the monsoon-circumglobal teleconnection hypothesis to changes of the selected rainfall region, showing the general robustness of the identified causal links. Both all-India rainfall (AIR) and western Himalayan foothills (WHF) rainfall behave consistently with the MT rainfall (see SI, Fig. S20-21). These similarities are likely due to the strong correlation that exists between the MT rainfall and AIR, being the MT rainfall one of the dominant features of the ISM intraseasonal variability (Krishnan et al., 2000). However, eastern Himalayan foothills (EHF) rainfall seems to behave differently, and does not show any connection with the updraft region identified by W1. This is likely to be related to the fact that the EHF region receives heavy rainfall amounts during the early stage of a break event and thus it is likely to be governed by different circulation pattern than those that govern the MT rainfall (Vellore et al., 2014). Nevertheless, also in this case, all other causal links in the CEN are retained.".

7. In all the network figures it would be useful to have the numbers indicating the path coefficient and auto-corr. path coefficient over the arrows and inside the circles, respectively. Absence of arrows indicates 0 path. Coefficient? Answer – We thank the reviewer for improving the layout of the figures. We have included the numbers for the path coefficient and auto-corr. path coefficient in all figure as suggested. Moreover, we

now specify in the caption of each figure that only significant links are shown.

8. Why linearity is a good framework here? Please cite works to justify this. Answer – DW2005 used linear correlations to define and analyse the circumglobal teleconnection pattern and its interaction with ISM. Moreover, Ding et al. 2011 further show that the circumglobal teleconnection pattern has a linear behaviour ... (EXPAND). Therefore we also assume in our analyses that individual links are to first order linear. We provide an additional supportive figure that shows that temperature and precipitation anomalies for high and low CGTI states behave close to linearly. See SI Fig. S4 and the related comment in the main text in lines 339-341: "Though these plots are obtained by plotting composites of weeks with CGTI > 1 CGTIstd minus composites of weeks with CGTI < - 1 CGTIstd, we also shown that these composites behave close to linearly when plotted separately (see SI, Fig. S4). Thus, this further supports the assumption that a linear framework is a reasonable choice in this context."

Minor comments:

The terms "mid-latitudes" and "extra-tropics" are used indistinctly. This is problematic and in "extra-tropics" the high-latitudes are also included. Please only use 1 term to avoid confusing the reader. – Answer - We thank the anonymous reviewer for pointing out this discrepancy, we now use the term "mid-latitudes" throughout the whole manuscript.

line 125: space missing ")algorithm" - Answer – we thank the reviewer for carefully reading the manuscript. We have corrected this typo.

line 129: space missing ":It should" - Answer – we thank the reviewer for carefully reading the manuscript. We have corrected this typo.

line 130: what is an actor? A variable? - Answer – We thank the anonymous reviewer for pointing out that the definition of actor was misplaced compared to its first appearance. We have now moved the first reference to the word "actor" in the place of its

first appearance. See lines 183-185: "A CEN detects and visualizes the causal relationships among a set of univariate time series of variables (here referred to as actors, (Kretschmer et al., 2016)."

line 133: what is near-linear? Define. - Answer – We have clarified the sentence, see lines 189-191: "Other important assumptions are stationarity of time series and near-linear interactions between actors, i.e. the selected actors should have a linear behaviour at least in a first order approximation."

line 134: PC algorithm means PC-MICI algorithm or other? Also the name is confusing as you used the term for Principal component before. – Answer – We thank the anonymous reviewer for pointing out the double meaning of this abbreviation. We have specified in the sentence that PC algorithm stands for Peter and Clark algorithm as correctly pointed out by the reviewer, see lines 192: "The PC-MCI algorithm is a two-step algorithm based on a modified version of the Peter and Clark (PC) algorithm (Runge et al., 2014; Spirtes et al., 2000)."

line 149: which are the 2 conditions? More, generally: which are the n-conditions? – Answer – We thank the anonymous reviewer for helping to improve the clarity of the methods section. We have improved the definition of the term condition, see lines 201-204: "Then, partial correlations between the ith actor and each element jth in Pi0 wher $i \neq j$, are calculated, conditioning first on one condition, i.e. the first kth actor in Pi0 that has the strongest correlation with the ith actor and $i \neq j \neq k$" and lines 151-152 "This process leads to a reduced set of parents Pi1. In the next step, the process is repeated but conditioning on two conditions, i.e. calculating the linear regression on a set of two actors, leading to a second set of parents Pi2.".

line 150: "parents contained in PЁξ{n}_i" is n the same as before? I guess not, change for m - Answer – We thank the reviewer for carefully reading the manuscript. We have corrected this typo.

line 161: why is only tau=1 selected? Justify, why not look at other taus? – Answer –
We have now included a Fig. that shows how the strength of the causal links decays with time. See comment #2.

Line 176-180: why is the correction needed? – Answer – The explanation for the corrected values is given in lines 180-182: "All p-values are corrected using the Benjamini and Hochberg false discovery rate (FDR) correction to address the variance inflation due to serial correlations (Benjamini and Hochberg, 1995; Benjamini and Yekutieli, 2001).".

Line 183: how is "circumglobal wave train" defined? – Answer – We meant here the circumglobal teleconnection pattern, we have corrected this phrase.

Line 186: why is the NAO influence included in this subsection? The title only talks about ISM and circumglobal wave train. – Answer – We have now changed the subtitle of this section to the following : "Causal testing of the two-way ISM-circumglobal teleconnection mechanism and the influence of NAO", see line ??.

fig 1: panel c: add in the title that it's over the MT region. Y-axis should be 10 to 10. Xaxis: it would be easier if it indicates the years. – Answer – We thank the reviewer for his/her suggestions. We have modified the first figure accordingly.

line 197: why start on the 2rd week? Justify and only show fig 1c starting at this time. – Answer – We have improved the explanation of why the first two time-steps need to be skipped, see line 238-240: "Skipping the two time steps of each year is necessary since the first two time slots allow detecting lagged relationships, and the PC-MCI algorithm requires to add twice the maximum time lag explored (here a maximum lag of 1 week is chosen)."

Line 199: justify why max lag 1 week is selected. –

Answer – We now show also lag -2 and -3 weeks for the main causal relationships in the SI (see previous comment #2). Moreover, the choice to present results for a one 1 week lag also reflect the interest on analysing the active and break phases

dynamics, which has a time period that goes from less than one up to three weeks (see introduction, lines 48-55).

line 243: typo "llink" – Answer – we thank the reviewer for carefully reading the manuscript. We have corrected this typo.

line 248-249: why the uncertainty in r? isn't it just the pattern correlation number? Please explain. – Answer – The uncertainty in r refers to the confidence interval of the correlation coefficient, to show whether it is significantly different from zero at a confidence level of 0.05. We now specify this the first time that the confidence interval appears (line 314).

Line 246-255: re-arrange paragraph to talk first about EOF1 and then about EOF2. Also, mention definition of Eurasian sector the first time it appears. – Answer – The apparent mixing is due to the fact that we choose to describe first the global spatial correlation and then the regional one. However, within this framework, EOF1 is always treated first. We now defined the Eurasian sector in line 316 as it first appears in the text.

Line 261-266: in Precipitation there is no signal over Western Europe. – Answer – We thank the reviewer for kindly pointing out this mistake, which we have now corrected.

line 269: Careful! How can you compare variables of different magnitude????!! you can't say the precipitation is weaker than temperature. – Answer – We intended to refer to the spatial extension of the precipitation anomalies, which are less spread than the temperature ones. We have now corrected the sentence as follow (lines 338-339): "Precipitation anomalies are more spatially confined than those found for 2m-temperature. However, a clearly defined wave pattern appears over the Eurasian sector. ".

Fig 2: CGTI has not been defined up to this point in the text. Why not show also lag + -1? seems important later. Why show temperature anomalies when it's not the

focus of the paper? Panels e and f: subtitles are misleading, indicate it's anomalies associated with extreme CGTI. – Answer – We have corrected the subtitles of Fig. 2, panels (e) and (f), as suggested by the reviewer. We decided to show temperature and precipitation anomalies since we consider important to know what the effect of the circumglobal teleconnection pattern phase is on other relevant atmospheric variables for summer weather in the mid-latitude, as this can have consequences for daily life in the affected regions. Moreover, we also double-check that high CGTI values are connected with positive rainfall anomalies over India. Lag -1 weeks results for both MT rainfall and CGTI are shown in Figure 4. Positive lags are not included in the current analysis.

Line 227-241: AT this point CGTI seems important to the paper, however it is not shown. I suggest to include its time-series. Also a justification on why such a small region is selected is needed. – Answer – As reported in lines 300-302, the CGTI regions is defined following DW2005. We define our CGTI in the same way, since they already show that small latitudinal variations do not affect the results (see Fig. 2 in DW2005). Following the reviewer's suggestion, we have now added a figure in the SI to show all time-series used in Fig. 7 (see SI, Fig. S19).

Line 316: OLR1 is not indicated in the figure 4b section 3.3: why add MJO in this section when it is about internal feedbacks? The sub-title is misleading – Answer – We thank the reviewer for pointing us to this discrepancy. Section 3.3 is meant to describe both tropical and internal variability. We have thus changed the subtitle accordingly to "Intraseasonal variability and tropical influence on the monsoon circulation".

fig 7: why is the network overlapped on a lon-lat map? No overlapping was done before. For example, why is EOF2 located over east Asia or W1 over west India? – Answer – This choice is made to help the reader to easily identify each actor with its corresponding geographical location. We have now included an explanation of this choice in lines 508-511: "We include the most important identified regions from both the tropics and the mid-latitudes together in a single CEN (Fig. 7) and plot the corresponding CEN

over a map to help the reader to associate each actor with its corresponding region (though in cases when the index is defined over all longitudes, such as EOF2, it is only possible to associate the actor with its average latitude)."

Line 446-447: you only mention mid-latitudes even though later tropics and internal feedbacks are analysed. – Answer – We have corrected the sentence accordingly to the reviewer's suggestion (lines 570-572): "In this study, we apply causal discovery algorithms to analyse the influence of global middle and upper tropospheric fields on the ISM rainfall and study the two-way causal links between the mid-latitude circulation and ISM rainfall, together with tropical drivers and ISM internal variability.".

Line 459: how does your results have implications for the interannual time-scales? I think this sentence is misleading. – Answer – It was meant "seasonal", as explained in lines 584-586. We thank the reviewer for pointing out this mistake, which has been now corrected.

Line 464-465: how does this pattern compare with the regression of MT precipitation on the Z200 field? Any substantial difference? – Answer – Following the reviewer's suggestion, we now provide the linear regression of the MT rainfall in the CGTI index, and show that the strongest relationship is found at lag -1 week, with the CGTI leading the Mt rainfall by one week: see SI Fig. S5 and the related comment in the main text in lines 342-345 "Moreover, when regressing the MT rainfall on the CGTI index at different lags (from lag 0 to lag -2 weeks), the strongest relationship between the CGTI and the MT rainfall is found at lag -1 week, with the CGTI leading a change in MT rainfall by one week (see SI, Fig. S5). This information also further supports the choice of analysing the relationships between these two variables at lag = -1 week.".

Line 486-488: is it possible to confirm this with your results? – Answer – Yes, at least partly. We now discuss the relationship between the BSISO, the initiation of active and break phases and the results obtained by our analysis in more details in the text (see comments to review #1). However, we do not analyse directly the excitation of Rossby

waves, and in this sense we cannot confirm this hypothesis within the scope of the present study.

Lines 502-508: why is the linear framework adequate for studying these mechanisms? Can you cite any modelling work implying linearity of these type of interactions? – Answer – see previous answer for comment #8.

Please also note the supplement to this comment:
https://www.earth-syst-dynam-discuss.net/esd-2019-42/esd-2019-42-AC2-supplement.pdf

**Supplement:**

[Figure]

**Figure S4. Temperature and precipitation anomalies related to high and low CGTI states.** Panels (a) and (b) show mean precipitation anomalies over the Northern Hemisphere during weeks with CGTI > 1 CGTI$_{s.d.}$ and weeks with CGTI < -1 CGTI$_{s.d.}$ respectively from NCEP CPC data and for the period 1979-2017. Panel (c) and (d) as for panel (a) and (b) but for temperature anomalies (from ERA-Interim reanalysis).

[Figure]

**Figure S5. Linear regression of the MT rainfall on the CGTI index.** Precipitation from CPC/NCEP for the period 1979-2917 linearly regressed on the CGTI index. Panel (a) show the regression coefficient (mm*day$^{-1}$*m$^{-1}$) for lag -2 (i.e. the CGTI leads the precipitation by 2 weeks). Panel (b) and panel (c) as for panel (a) but for lags -1 and lag 0 respectively.

135

[Figure]

**Figure S11.** Time evolution of path coefficients for a CEN built with CGTI, W1, MJO2 and MT (over the period 1979-2017) from lag -1 up to lag -3 weeks. Circles denote significant values at pval < 0.5, while nonsignificant values are plotted with triangles.

[Figure]

175

**Figure S12. CGTI and MT rainfall at monthly time scale.** Panel (a): correlation between monthly  MT rainfall (lag = 0) and Z200 (lag = -1 month). Panel (b): as panel (a) but with monthly MT rainfall (lag = 0) and Z200 (lag = 0). Panel (c) as panel (a) but for MT rainfall and W. Panel (d): correlation between CGTI and Z200 (both at lag = 0), which forms the circumglobal teleconnection pattern. In panels (c) and (d), correlation coefficients and anomalies with a *p*-value of $p < 0.05$ (accounting for the effect of serial correlations) are shown by black

180     contours.

[Figure]

225  **Figure S20. AIR, WHF and EHF rainfall.** Panel (a) shows all-India rainfall (AIR) for CPC-NCEP data. Panel (b) shows western Himalayan foothills (WHF, defined over 26°-35°N and 70°-83°E) and eastern Himalayan foothills (EHF, defined over 20°-30°N and 87°-97°E) contoured by a black box. Panel (c) shows the time series for AIR averaged over the whole country for the period 1979-2017. Panel (d) and (e) as panel (c) but for WHF and EHF rainfall respectively. See main text for the description of the results.

[Figure]

**Figure S21. CEN for AIR, WHF and EHF rainfall.** Panel (a) shows a CEN as for Fig. 7 in the main text but for western Himalayan foothills (WHF) rainfall from CPC-NCEP data. Panel (b) and (c) as panel (a) but for eastern Himalayan foothills (EHF) rainfall and all-India rainfall (AIR) respectively. See main text for the description of the results.

230

| Link | Link strength (CE) | Simple correlation |
|---|---|---|
| $\beta_{W1\to MT}$ | 0.54 | 0.55 |
| $\beta_{MJO2\to W1}$ | 0.49 | 0.45 |
| $\beta_{CGTI\to MT}$ | 0.18 | 0.26 |
| $\beta_{W1\to MJO2}$ | -0.39 | 0.17 |
| $\beta_{EOF2\to NAO}$ | 0.12 | -0.01 |
| $\beta_{NAO\to CGTI}$ | 0.17 | 0.12 |
| $\beta_{W1\to CGTI}$ | 0.16 | 0.2 |
| $\beta_{EOF2\to CGTI}$ | 0.14 | 0.16 |
| $\beta_{CGTI\to EOF2}$ | 0.07 | 0.15 |
| $\beta_{CGTI\to W1}$ | 0.09 | 0.15 |
| $\beta_{MJO2\to W1}*\beta_{W1\to MT}$ | 0.49*0.54 = 0.26 | 0.38 |
| $\beta_{CGTI\to W1}*\beta_{W1\to MT}$ | 0.09*0.54 = 0.05 | 0.26 |
| $\beta_{NAO\to CGTI}(\beta_{CGTI\to MT} + \beta_{CGTI\to W1}*\beta_{W1\to MT})$ | 0.17*(0.18+0.05) = 0.04 | 0.04 |

240   **Table S1. Causal effect (CE) values.** CE values for links presented in Fig. 7 in the main text.

|  | MT rainfall | NAO | CGTI | EOF2 | W1 | MJO2 |
|---|---|---|---|---|---|---|
| ACE | 0.0 | 0.033 | 0.066 | 0.051 | 0.217 | 0.098 |
| ACS | 0.143 | 0.023 | 0.093 | 0.013 | 0.116 | 0.077 |

**Table S2. Average causal effect (ACE) and average casual susceptibility (ACS).** ACE and ACS for actors presented in Fig. 7 in the main text.

245